# A Unified View of Differentially Private Deep Generative Modeling

**Dingfan Chen**[*]                                                                          *dingfan.chen@cispa.de*
*CISPA Helmholtz Center for Information Security*

**Raouf Kerkouche**                                                                          *raouf.kerkouche@cispa.de*
*CISPA Helmholtz Center for Information Security*

**Mario Fritz**                                                                              *fritz@cispa.de*
*CISPA Helmholtz Center for Information Security*

**Reviewed on OpenReview:** *https://openreview.net/forum?id=YgmBD2c9qX*

## Abstract

The availability of rich and vast data sources has greatly advanced machine learning applications in various domains. However, data with privacy concerns comes with stringent regulations that frequently prohibit data access and data sharing. Overcoming these obstacles in compliance with privacy considerations is key for technological progress in many real-world application scenarios that involve sensitive data. Differentially private (DP) data publishing provides a compelling solution, where only a sanitized form of the data is publicly released, enabling privacy-preserving downstream analysis and reproducible research in sensitive domains. In recent years, various approaches have been proposed for achieving privacy-preserving high-dimensional data generation by private training on top of deep neural networks. In this paper, we present a novel unified view that systematizes these approaches. Our view provides a joint design space for systematically deriving methods that cater to different use cases. We then discuss the strengths, limitations, and inherent correlations between different approaches, aiming to shed light on crucial aspects and inspire future research. We conclude by presenting potential paths forward for the field of DP data generation, with the aim of steering the community toward making the next important steps in advancing privacy-preserving learning.

## 1 Introduction

Data sharing is crucial for the growth of machine learning applications across various domains. However, in many application scenarios, data sharing is prohibited due to the private nature of data (e.g., individual data from mobile devices, medical treatments, and banking records) and associated stringent regulations, such as the General Data Protection Regulation (GDPR) and the American Data Privacy Protection Act (ADPPA), which largely hinders technological progress in sensitive areas. Fortunately, differentially private (DP) data publishing (Dwork, 2008; Dwork et al., 2009; Fung et al., 2010) provides a compelling solution, where only a sanitized form of the data, with rigorous privacy guarantees, is made publicly available. Such sanitized synthetic data can be leveraged as a surrogate for real data, enabling downstream statistical analysis using established analytic tools, and can be shared openly with the research community, promoting reproducible research and technological advancement in sensitive domains.

Traditionally, the sanitization algorithms are designed for capturing low-dimensional statistical characteristics and target at specific downstream tasks (e.g., answering linear queries (Roth & Roughgarden, 2010; Hardt & Rothblum, 2010; Blum et al., 2013; Vietri et al., 2020)), which are hardly generalizable to unanticipated tasks involving high-dimensional data with complex distributions. On the other hand, the latest research, inspired by the recent successes of deep generative models in learning high-dimensional representations, applies deep

---

[*]Correspondence to: Dingfan Chen (dingfan.chen@cispa.de)

generative models as the foundation of the generation algorithm. This line of approaches, as demonstrated in recent studies (Cao et al., 2021; Chen et al., 2020a; Xie et al., 2018; Yoon et al., 2019; Beaulieu-Jones et al., 2017; Ghalebikesabi et al., 2023), have shown promising results in sanitizing high-dimensional samples for general purposes.

Towards designing models that are better compatible with the privacy target, recent research typically customizes the training objective for privacy-centric scenarios (Cao et al., 2021; Harder et al., 2021; Chen et al., 2020a; Long et al., 2021), all building on top of a foundational generic generator framework. However, research is fragmented as contributions have been made in different domains, different modeling paradigms, different metric and discriminator choices, and different data modalities. So far, a unified view of private generative models is notably missing in the literature, despite its potential to consolidate the design space for systematic exploration of innovative architectures and leveraging strengths across diverse modeling frameworks.

In this paper, we pioneer in providing a comprehensive framework and a unified perspective on existing approaches for differentially private deep generative modeling. Our innovative framework, complemented by an insightful taxonomy, effectively encapsulates approaches from existing literature, categorizing them according to the intrinsic differences in their underlying privacy barriers. We thoroughly assess each category's characteristics, emphasizing crucial points relevant for privacy analysis, and discuss their inherent strengths and weaknesses, with the aim of laying a foundation that supports seamless transition into potential future research.

Moreover, we present a thorough introduction to the key concepts of DP and generative modeling. We highlight the key considerations that should be accounted for when developing DP generative models to ensure results comparable, error-free results. Furthermore, we introduce a taxonomy of existing representative types of deep generative models, classifying them based on the distinctive privacy challenges present during DP training. This introduction aims to equip researchers and practitioners with a systematic approach for the design and implementation of future privacy-preserving data generation techniques.

Lastly, we discuss open issues and potential future directions in the broader field of developing DP generation methods. Our objective is not limited to reviewing existing techniques, but also aims to equip readers with a systematic perspective for devising new approaches or refining existing ones. This work is thoughtfully written to serve diverse audiences, with an effort of providing practitioners with a comprehensive overview of the recent advancements, while aiding experts in reassessing existing strategies and designing innovative solutions for privacy-preserving generative modeling.

## 2 Preliminaries of Differential Privacy

**Setting.** In this paper, we focus on the standard *central* model of DP, which is commonly agreed upon by all the approaches referenced herein. In this model, a trusted party or server is responsible for managing all data points, executing DP algorithms, and producing sanitized data that conforms to privacy constraints. This sanitized data, generated from the implemented DP algorithms, can be later shared with untrusted parties or released to the public while ensuring strict privacy guarantees. It is noteworthy that although approaches based on local DP may seem to generate a form of synthetic data—where users typically modify their own data due to distrust in the central server and a desire to conceal private information—these methods are fundamentally distinct from the ones explored in this work due to differing threat models and the resulting privacy implications.

**Definition 2.1** (($\varepsilon, \delta$)-DP (Dwork, 2008)). A randomized mechanism $\mathcal{M}$ with range $\mathcal{R}$ is ($\varepsilon, \delta$)-DP, if

$$\Pr[\mathcal{M}(\mathcal{D}) \in \mathcal{O}] \leq e^{\varepsilon} \cdot \Pr[\mathcal{M}(\mathcal{D}') \in \mathcal{O}] + \delta$$

holds for any subset of outputs $\mathcal{O} \subseteq \mathcal{R}$ and for any adjacent datasets $\mathcal{D}$ and $\mathcal{D}'$, where $\mathcal{D}$ and $\mathcal{D}'$ differ from each other with only one training example. $\varepsilon$ is the upper bound of privacy loss, and $\delta$ is the probability of breaching DP constraints. Smaller values of both $\varepsilon$ and $\delta$ translate to stronger DP guarantees and better privacy protection. Typically, $\mathcal{M}$ refers to the training algorithm of a generative model. DP ensures that inferring the presence of an individual in the private dataset—by observing the trained generative models

$\mathcal{M}(\mathcal{D})$—is challenging, with $\mathcal{D}$ being the original private dataset. This same level of guarantee also holds when the attacker observes the samples generated by the trained generative models (i.e., the sanitized dataset) due to the post-processing theorem (Theorem 2.1).

**Privacy notion.** There are two widely used definitions for adjacent datasets in existing works of DP data generation, which result in different DP notions: the "replace-one" and the "add-or-remove one" notions:

- **Replace-one**: adjacent datasets are formed by *replacing* one data sample, i.e., $\mathcal{D}' \cup \{x'\} = \mathcal{D} \cup \{x\}$ for some $x$ and $x'$. This is sometimes referred to *bounded-DP* in literature.
- **Add-or-remove-one**: adjacent datasets are constructed by *adding* or *removing* one data sample, i.e., $\mathcal{D}' = \mathcal{D} \cup \{x\}$ for some $x$ (or vice versa).

It is crucial to understand that different notions of DP may not provide equivalent privacy guarantees even under identical $(\varepsilon, \delta)$ values, potentially leading to slight differences in comparisons when algorithms are developed under varying privacy notions, a sentiment also noted in Ponomareva et al. (2023). Specifically, the "replacement" operation in the bounded-DP notion can be understood as executing two edits: removing one data point $x$ and adding another $x'$. This suggests that the *replace-one* notion may be nested within the *add-or-remove-one* notion, and a naive transformation would result in a $(2\varepsilon, \delta)$-DP algorithm under the *replace-one* notion from an algorithm that was $(\varepsilon, \delta)$-DP under the *add-or-remove-one* notion. To minimize potential confusion and promote fair comparisons, we emphasize that future researchers should clearly specify the chosen notion in their work. Moreover, we encourage future research to include a privacy analysis for both notions, if technically feasible.

Privacy-preserving data generation is building on top of the closedness of DP under post-processing: if a generative model is trained under a $(\varepsilon, \delta)$-DP mechanism, releasing a sanitized dataset generated by the model (for conducting downstream analysis tasks) will also be privacy-preserving, with the privacy cost bounded by $\varepsilon$ (and $\delta$).

**Theorem 2.1** (Post-processing (Dwork et al., 2014))**.** If $\mathcal{M}$ satisfies $(\varepsilon, \delta)$-DP, $F \circ \mathcal{M}$ will satisfy $(\varepsilon, \delta)$-DP for any data-independent function $F$ with $\circ$ denoting the composition operator.

While $(\varepsilon, \delta)$-DP provides an intuitive understanding of the mechanism's overall privacy guarantee, dealing with composition is more convenient under the notion of Rényi Differential Privacy (RDP). Existing approaches typically use RDP to aggregate privacy costs across a series of mechanisms (such as multiple DP gradient descent steps during generative model training) and then convert to the $(\varepsilon, \delta)$-DP notion at the end (See Appendix D). The formal definitions and the corresponding theorems are listed below.

**Definition 2.2** (Rényi Differential Privacy (RDP) (Mironov, 2017))**.** A randomized mechanism $\mathcal{M}$ is $(\alpha, \rho)$-RDP with order $\alpha$, if

$$D_\alpha(\mathcal{M}(\mathcal{D}) \| \mathcal{M}(\mathcal{D}')) = \frac{1}{\alpha - 1} \log \mathbb{E}_{t \sim \mathcal{M}(\mathcal{D})} \left[ \left( \frac{\Pr[\mathcal{M}(\mathcal{D}) = t]}{\Pr[\mathcal{M}(\mathcal{D}') = t]} \right)^\alpha \right] \leq \rho$$

holds for any adjacent datasets $\mathcal{D}$ and $\mathcal{D}'$, where $D_\alpha(P \| Q) = \frac{1}{\alpha - 1} \log \mathbb{E}_{t \sim Q}[(P(t)/Q(t))^\alpha]$ denotes the Rényi divergence.

**Theorem 2.2** (Composition (Mironov, 2017))**.** For a sequence of mechanisms $\mathcal{M}_1, ..., \mathcal{M}_k$ s.t. $\mathcal{M}_i$ is $(\alpha, \rho_i)$-RDP $\forall i$, the composition $\mathcal{M}_1 \circ ... \circ \mathcal{M}_k$ is $(\alpha, \sum_i \rho_i)$-RDP.

**Theorem 2.3** (From RDP to $(\varepsilon, \delta)$-DP (Balle et al., 2020))**.** If a randomized mechanism $\mathcal{M}$ is $(\alpha, \rho)$-RDP, then $\mathcal{M}$ is also $\left( \rho + \log((\alpha - 1)/\alpha) - (\log \delta + \log \alpha)/(\alpha - 1), \delta \right)$-DP for any $0 < \delta < 1$.

In literature, achieving DP typically involves adding calibrated random noise, with scale proportional to the sensitivity value (Definition 2.3), to the private dataset's associated quantity to conceal individual influence. A notable instance of this practice can be formularized as the Gaussian Mechanism, as defined below.

**Definition 2.3** (Sensitivity)**.** The (global) $\ell_p$-sensitivity for a function $f : X \to \mathbb{R}^d$ that outputs $d$-dimensional vectors is defined as:

$$\Delta_f^p = \max_{\mathcal{D}, \mathcal{D}'} \| f(\mathcal{D}) - f(\mathcal{D}') \|_p \tag{1}$$

over all adjacent datasets $\mathcal{D}$ and $\mathcal{D}'$. The sensitivity characterizes the maximum influence (measured by $\ell_p$ norm) of one individual datapoint on the function's output. When dealing with matrix and tensor outputs, the $\ell_p$ norm is computed over the vectors that result from flattening the matrices and tensors into vectors.

**Definition 2.4** (Gaussian Mechanism (Dwork et al., 2014)). Let $f : X \to \mathbb{R}^d$ be an arbitrary $d$-dimensional function with $\ell_2$-sensitivity $\Delta_f^2$. The Gaussian Mechanism $\mathcal{M}_\sigma$, parameterized by $\sigma$, adds noise into the output, i.e.,

$$\mathcal{M}_\sigma(\boldsymbol{x}) = f(\boldsymbol{x}) + \mathcal{N}(0, \sigma^2 \boldsymbol{I}). \tag{2}$$

$\mathcal{M}_\sigma$ is $(\varepsilon, \delta)$-DP for $\sigma \geq \sqrt{2 \ln{(1.25/\delta)}}\Delta_f^2/\varepsilon$ and $(\alpha, \frac{\alpha(\Delta_f^2)^2}{2\sigma^2})$-RDP.

## 2.1 Training Deep Learning Models with DP

Additionally, we present the most prominent frameworks for training deep learning models with DP guarantees: Differentially Private Stochastic Gradient Descent (DP-SGD) in Section 2.1.1 and Private Aggregation of Teacher Ensembles (PATE) in Section 2.1.2.

### 2.1.1 Differenetially Private Stochastic Gradient Descent (DP-SGD)

DP-SGD (Abadi et al., 2016) is an adaptation of the standard SGD algorithm that injects calibrated random Gaussian noise into the gradients during the optimization process, which ensures DP due to the Gaussian mechanism. The algorithm consists of the following steps:

1. Compute the per-example gradients for a mini-batch of training examples.
2. Clip the gradients to bound their $\ell_2$-norm (i.e., $\ell_2$-sensitivity) to ensure that the influence of any individual training example is limited.
3. Add Gaussian noise to the aggregated clipped gradients to introduce the required randomness for DP guarantees.
4. Update the model parameters using the noisy gradients.

The privacy guarantees provided by DP-SGD are determined by the choice of noise multiplier (which defines the standard deviation of the Gaussian noise by multiplying it with the sensitivity), the mini-batch sampling ratio, and the total number of optimization steps. The overall privacy guarantee can be calculated using the composition rule, which accounts for the cumulative privacy loss over multiple iterations of the algorithm. By default, DP-SGD adopts the *add-or-remove-one* notion, leading to a sensitivity value equal to the gradient clipping bound (see Appendix C).

### 2.1.2 Private Aggregation of Teacher Ensembles (PATE)

The PATE framework (Papernot et al., 2017; 2018) consists of two main components: an ensemble of teacher models and a student model. The training process begins with the partitioning of sensitive data into multiple disjoint subsets. Each subset is then used to train a teacher model independently (and non-privately), limiting the effect of each individual training sample to influence only one teacher model. To train a DP student model, a public dataset with similar characteristics to the sensitive data is used. During the training process, the student model queries the ensemble of teacher models for predictions on the public dataset. The teacher models' predictions are then aggregated using a DP voting mechanism, which adds noise to the aggregated votes to ensure privacy. The student model subsequently learns from the noisy aggregated predictions, leveraging the collective knowledge of the teacher models while preserving the privacy of the original training data.

The sensitivity of PATE is measured as the maximum change in label counts for teacher models' predictions between neighboring datasets. Given $m$ teacher models, $c$ label classes, the counts for class $j$ is defined by the number of teachers that assign class $j$ to a query input $\bar{\boldsymbol{x}}$, i.e., $n_j(\bar{\boldsymbol{x}}) = |i : i \in [m], f_i(\bar{\boldsymbol{x}}) = j|$ for $j \in [c]$, where $f_i$ denotes the $i$-th teacher model. Changing a single data point (whether by replacing, adding, or removing) will at most affect one data partition and, consequently, the prediction for one teacher trained on

the altered partition, increasing the counts by 1 for one class and decreasing the counts by 1 for another class. This results in a global sensitivity equal to $\Delta^2_{(n_1,...,n_c)} = \sqrt{2}$ for both the *replace-one* and *add-or-remove-one* notion (see Appendix C). To reduce privacy consumption, PATE is associated with a data-dependent privacy accountant method to exploit the fact that when teachers have a large agreement, the privacy cost is usually much smaller than the data-independent bound would suggest. Moreover, Papernot et al. (2018) suggest private threshold checking for queries to only use teacher predictions with high consensus for training the student model. Notably, to obtain comparable results to approaches with data-independent privacy costs, extra sanitization via smooth sensitivity analysis is required.

## 2.2 Important Notes for Deploying DP Models

The development of DP models necessitate a thorough examination to ensure their correctness for providing a fair comparison of research progress and maintaining public trust in DP methodologies. We present below a series of critical questions that serve as fundamental sanity checks when developing DP models. This enables researchers to rapidly identify and rule out approaches that are incompatible with DP, thereby optimizing their research efforts towards innovation in this domain.

- **What will be released to the public and accessible to potential adversaries?** The most critical question is to determine which components (e.g., model modules, data statistics, intermediate results, etc.) will be made public and, as a result, could be accessible to potential adversaries. This corresponds to the assumed ***threat model*** and establishes the essential concept of a ***privacy barrier***, which separates components accessible to potential attackers from those that are not.

  All components within the attacker-accessible domain must be provided with DP guarantees. One common oversight is neglecting certain data-related intermediate statistics utilized during the model's training phase. These statistics might constitute only a minor aspect of the entire process, or their existence might be implicit, given that they are incorporated into other quantities. Nevertheless, failing to implement DP sanitization for these aspects can undermine the intended DP protection for the outcomes, e.g., the trained model may no longer adhere to DP standards.

  For instance, when pre-processing is required for the usage of a DP model, an additional privacy budget should be allocated for exposing related statistics such as the dataset's mean and standard deviation (Tramer & Boneh, 2021). From a research standpoint, innovations may involve carefully designing DP mechanisms that apply DP constraints only to components accessible by attackers, while other components can be trained or computed non-privately to maintain high utility. A concrete example includes training a discriminator non-privately and withholding it by the model owner in deploying DP generative adversarial networks (see Section 4.3) while only privatizing the generator's training and releasing it to the public with a dedicated DP mechanism.

- **What is the adopted privacy notion and granularity?** While DP asserts that an algorithm's output remains largely unchanged when a single database entry is modified, the definition of a "single entry" can vary considerably (reflecting the concept of ***granularity***), and the way to modify the single entry can also be different (embodying the ***privacy notion***). Thus, the claims of DP necessitate an unambiguous declaration of the sense and level at which privacy is being promised. As discussed in the previous section, the distinction in privacy notion is universally crucial in the design of DP mechanisms. On the other hand, the granularity becomes particularly relevant when handling data modalities that exhibit relatively less structural representations, such as graphs and text. For instance, training DP (generative) language models that provide guarantees at different levels (tokens, sentences, or documents) will lead to substantial differences in the complexity and the application scenarios.

- **What constitutes the sensitivity analysis?** Sensitivity analysis demands rigorous attention, focusing on two primary aspects. The first consideration calls for a clear statement of the ***sensitivity type*** in use, e.g., global, local, and smooth sensitivity. Notably, techniques predicated on local and smooth sensitivity are generally not directly comparable to those depending on the global sensitivity. Second, determining the sensitivity bound during the training of a generative model that consists of more than one trainable module may be challenging, as discussed in Section 4.4, which necessitates a meticulous analysis to ensure the correctness of the privacy cost computation.

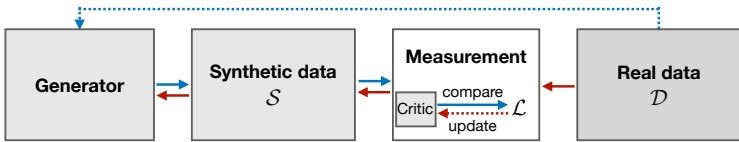

**Figure 1:** Overview of training pipeline of generative models. (**Blue arrow**: forward pass; **Red arrow**: backward pass; Dashed arrows indicate optional processes that may not be present in all generative models.)

## 3   Preliminaries of Generative Models

In this section, we present a comprehensive overview of representative generative models, with the aim to develop a clear understanding of the essential operations required to achieve DP across different types of generative models, as well as to demonstrate the fundamental differences in their compatibility with private training.

### 3.1   Overview & Taxonomy

Given real data samples $\boldsymbol{x}$ from a dataset of interest, the goal of a generative model is to learn and capture the characteristics of its true underlying distribution $p(\boldsymbol{x})$ and subsequently allows the model to generate new samples from the learned distribution. At a high-level of abstraction, the training pipeline of generative models can be depicted as the diagram in Figure 1. The "Measurement" block in the diagram summarizes the general process of comparing the synthetic and real data distributions using a "critic", which yields a loss term $\mathcal{L}$ that quantifies the similarity between the two. This loss term then acts as the training objective for the generator, with the update signal computed and then backpropagated to adjust the generator's parameters and improve its ability to generate realistic samples.

Furthermore, the diagram outlines two optional processes (indicated by dashed arrows), that are involved in some generative models but not all. The first optional process involves guiding the training of the generator by feeding (quantities derived from) real data as inputs, which enables the explicit maximum likelihood computation and categorizes the models into two types: *implicit density* and *explicit density*. The second optional process involves updating the critic to better capture the underlying structure of the data and more accurately reflect the similarity between the distributions. This distinction highlights the usage of either *static (data-independent)* or *learnable (data-dependent)* features for the critic function within implicit density models.

We present a taxonomy of existing representative types of generative models whose private training has been realized in literature in Figure 2. We examine the following tiers in the taxonomy trees that exert significant influence on the application scenarios and the design of corresponding private training algorithms:

- *Explicit* vs. *Implicit* Density Models
- *Learnable* vs. *Static* Critics
- *Distribution-wise* vs. *Point-wise* Optimization
- *Tractable* vs. *Approximate* Density

***Explicit* vs. *Implicit* Density Models.**   Existing generative models can be divided into two main categories: *explicit density models* define an explicit density function $p_{\text{model}}(\boldsymbol{x}; \boldsymbol{\theta})$, while *implicit density models* learn a mapping that generates samples by transforming an easy-to-sample random variable, without explicitly defining a density function.

These distinctions in modeling design result in different paradigms during the training phase, particularly in how real data samples are used (or accessed) in the process. *Explicit* density models typically use real data samples as inputs to the generator and also for measurement (as demonstrated in Figure 1), thereby enabling the tractable computation or approximation of the data likelihood objective. In contrast, *implicit* density models necessitate real data samples solely for the purpose of distribution comparison measurements.

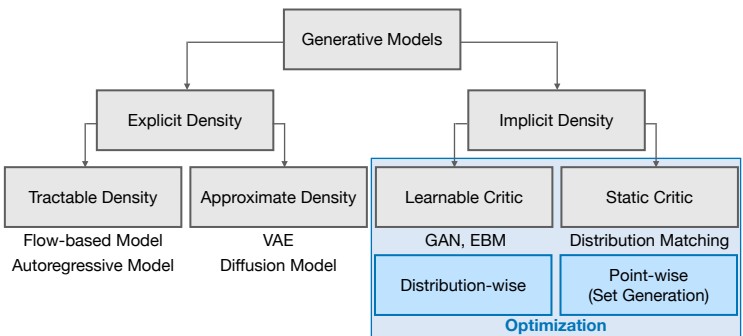

**Figure 2:** Overview of different deep generative models.

This distinction demarcates potential privacy barriers for these two types of models during DP model training. In the context of implicit models, it is sufficient to privatize the single access point to the real data (Section 4.1-4.3). However, when dealing with training private *explicit* density models, it becomes essential to apply DP mechanisms that take both access points into account.

***Learnable* vs. *Static* Critics.** The training of generative models necessitates a "critic" to assess the distance between the real and generated distributions, which then builds up the training objectives for optimizing the generator. Specifically for *implicit* density models, the use of different types of critics could potentially influence the placement of privacy barrier when training DP models (Section 4.1- 4.2).

Within this framework, the critics may exist in two primary forms, namely *learnable* and *static* (data-independent) variants. The distinction between the two lies in whether the critic itself is a parameterized function that undergoes updates during the training of generative models (*learnable*), or a data-independent function that remains static during the training process (*static*).

We do not further differentiate for explicit density models as they typically employ simple, data-independent critic such as $\ell_1$ and $\ell_2$ losses. Meanwhile, in contrast to implicit models, varying the critics in explicit models typically does not alter the privacy barrier in DP training. This is due to the constraint imposed by multiple access to real data in training of explicit models, which restricts the flexibility in positioning the privacy barriers.

***Distribution-wise* vs. *Point-wise* Optimization.** Generative models are designed to be stochastic and capable of producing a distribution of data. This is achieved by supplying the generator with random inputs (i.e., latent variables), stochastically drawn from a simple distribution, such as the standard Gaussian. The optimization process generally proceeds through mini-batches, essentially serving as point-wise approximations. Through substantial number of update steps that involve various random latent variable inputs, the model is trained to generalize over new random variables during the generation phase, enabling a smooth transition from a *point-wise* approximation to the *distribution-wise* objective.

However, certain contexts may not necessitate the stochasticity nature in these models. Instead, there might be an intentional focus on generating a small set of representative samples, a notion that resonates with the "coreset" concept. This could involve optimizing the model over a limited, fixed set of random inputs rather than the entire domain. We label this as *point-wise optimization* to distinguish it from the default *distribution-wise optimization* used in training conventional generative models.

Recent studies have revealed intriguing advantages of merging insights from both these strategies, particularly in the realm of private learning. For instance, the point-wise optimization method exhibits remarkable compatibility with private learning primarily arises from the fact that *point-wise* optimization is generally less challenging in comparison to the *distribution-wise* training that requires generalization, which generally improves model convergence, and consequently enhances privacy. However, this point-wise approach has its limitations. Unlike distribution-wise training, it does not inherently support generalization over new latent

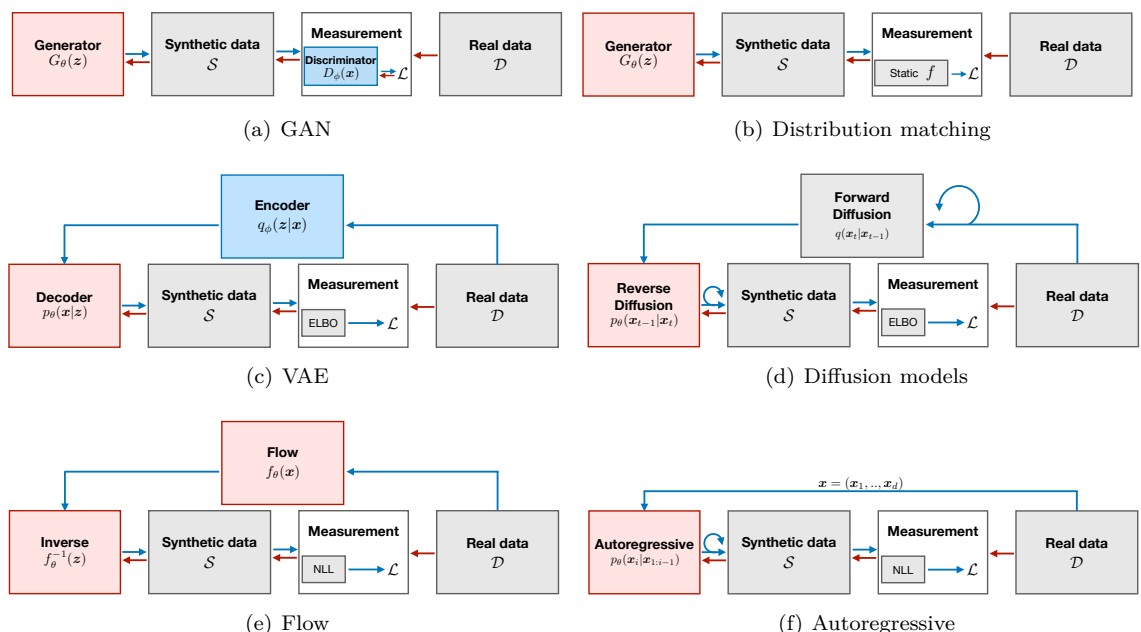

**Figure 3:** Diagram illustrating training process in generative models. **Blue arrow**: forward pass; **Red arrow**: backward pass.

code inputs. This may restrict the stochastic sampling of new synthetic samples during inference. As a result, there is a trade-off between the flexibility of use in downstream applications and improved privacy guarantees.

We do not expressly differentiate between potential optimization strategies for explicit density models within our taxonomy in Figure 2, as such distinction is not obvious in the context of explicit density models. In these models, the latent space is typically formulated through a transformation of the distribution within the data space. This transformation process in turn complicates the control of stochasticity throughout the training phase and diminishes the applicability of point-wise optimization.

***Tractable*** **vs.** ***Approximate*** **Density.** For models defining explicit density, a key distinguishing factor that shows practical relevance pertains to whether they allow exact likelihood computations. These models can broadly be categorized into two types: *tractable* density and *approximate* density models. The classification primarily stems from the model structural designs, which either enable tractable density inference or fall within the realm of approximate density.

Existing studies have demonstrated encouraging results when conducting DP training on both types of models. Intriguingly, the DP training mechanisms appear to exhibit minor distinctions when applied to these two different categories. On an optimistic note, such results implies that it might be feasible to attain tractable likelihood computations with a DP guarantee without considerable effort. However, it remains unclear as to whether the difference in model designs will systematically influence their compatibility with DP training.

## 3.2 Representative Models

We provide an illustration of the operational flow of representative generative models in Figure 3. As demonstrated, existing representative generative models can be effectively encapsulated within our unified framework shown in Figure 1. We proceed to briefly discuss the key characteristics of each type of generative models and their relation to potential implementations for DP training in this subsection.

### 3.2.1 Implicit Density Models

As a canonical example of implicit density model, *Generative Adversarial Network (GAN)* (Goodfellow et al., 2014) employs a generator, $G_{\boldsymbol{\theta}}$ (parametrized by $\boldsymbol{\theta}$), to learn the data distribution with the aid of a discriminator $D_{\boldsymbol{\phi}}$ (parametrized by $\boldsymbol{\phi}$) trained jointly in an adversarial manner, obviating the need for explicit density definition. The generator's functionality is enabled by inputting random latent variables, $\boldsymbol{z}$, drawn from simple distributions such as a standard Gaussian, and mapping these random inputs to the data space. Concurrently, the discriminator is provided with both synthetic and real samples and its training objective is to differentiate between the two. Throughout the training process, the generator and the discriminator compete and evolve, enabling the generator to create realistic samples that can deceive the discriminator, while the discriminator enhances its ability to distinguish between real and fake samples. The original GAN training objective can be interpreted as optimizing the generator to produce synthetic data that minimizes the Jensen-Shannon (JS) divergence between the synthetic and real data distributions. This idea has been expanded in various GAN training objective extensions explored in the literature. For instance, variants have been proposed based on generalizations to any f-divergence (Nowozin et al., 2016), Wasserstein distance (Arjovsky et al., 2017; Gulrajani et al., 2017), maximum mean discrepancy (MMD) (Bińkowski et al., 2018; Li et al., 2017), and Sinkhorn distance (Genevay et al., 2018).

Of particular interest to DP training is the observation that many of these divergence metrics can be approximated without requiring the training of a discriminator network. This has led to recent research in private generative models, which use a static function as the critic instead of a discriminator network. While such approaches might fall short in standard (non-private) generative modeling due to a lower expressive power compared to using learnable critic (that is adaptable to large data with diverse properties), they are highly competitive in DP training, as a static critic can effectively speed up convergence, thereby improving privacy guarantees.

In the case of implicit density models, the generator's interaction with the private dataset is typically indirect (only via the backward pass), meaning that there exists no direct link between the data source, as illustrated in the accompanying diagrams (Figure 3). This configuration presents an opportunity to strategically position the privacy barrier anywhere along the backpropagation path where the generator retrieves signals from the real data, facilitating an improved signal-to-noise ratio or simplified implementation. A more comprehensive understanding is presented in Section 4.1-4.3.

### 3.2.2 Explicit Density Models

Several prominent explicit density models have been developed in literature, each with distinct characteristics:

- The *Variational Autoencoder (VAE)* (Kingma & Welling, 2014) is trained to maximize the Evidence Lower Bound (ELBO), a lower bound of the log-likelihood, which typically simplifies to $\ell_1/\ell_2$ losses on the data sample and its reconstruction under standard Laplacian/Gaussian noise modeling assumptions. The model comprises trainable encoder and decoder modules. Encoding is conducted through the encoder $q_{\boldsymbol{\phi}}$, which maps observed data to its corresponding latent variables, denoted as $\boldsymbol{x} \overset{q_{\boldsymbol{\phi}}}{\to} \boldsymbol{z}$. The dimensions of these latent variables are typically smaller than the data dimension $d$, embodying the concept of an information bottleneck (Tishby et al., 1999; Shwartz-Ziv & Tishby, 2017). The decoder module is responsible for data reconstruction or generation, i.e., $\boldsymbol{z} \overset{p_{\boldsymbol{\theta}}}{\to} \boldsymbol{x}$. Additionally, VAE imposes regularization on the latent distributions to match the pre-defined prior, thereby enabling the generation of valid novel samples during inference.

- *Diffusion models* (Sohl-Dickstein et al., 2015; Song & Ermon, 2019; Ho et al., 2020) operate similarly to VAEs in terms of maximizing the ELBO. However, instead of using a trainable encoder to map data to latent variables, diffusion models transform the data iteratively through a linear Gaussian operation, represented as $\boldsymbol{x} \overset{q}{\to} ... \overset{q}{\to} \boldsymbol{x}_{t-1} \overset{q}{\to} \boldsymbol{x}_t \overset{q}{\to} \boldsymbol{x}_T$. This procedure causes the latent variables at the final step $\boldsymbol{x}_T$ to form a standard Gaussian distribution and maintain the same dimensionality as the data. The generation process is executed by reversing the diffusion operation, which means iteratively applying $p_{\boldsymbol{\theta}}(\boldsymbol{x}_{t-1}|\boldsymbol{x}_t)$ for all time steps $t \in [T]$. The trainable component of diffusion models resides in the reverse diffusion process, while the forward process is pre-defined and does not require training.

- *Flow-based* models (Rezende & Mohamed, 2015; Kingma & Dhariwal, 2018), in contrast, minimize the Negative Log-Likelihood (NLL) directly. Uniquely, flow-based models employ the same invertible model for both encoding ($\boldsymbol{x} \xrightarrow{f_{\boldsymbol{\theta}}} \boldsymbol{z}$) and generation ($\boldsymbol{z} \xrightarrow{f_{\boldsymbol{\theta}}^{-1}} \boldsymbol{x}$), by executing either the flow or its inverse. Due to the invertibility demanded by the model construction, the dimensions of the latent variables $\boldsymbol{z}$ are identical to those of the data.

- *Autoregressive models* (Larochelle & Murray, 2011; Oord et al., 2016; Van Den Oord et al., 2016; Van den Oord et al., 2016), as another instance of model with tractable density, are also designed to minimize the NLL. Unlike some other models, they accomplish this without the need for explicit latent variables or an encoding mechanism. Instead, these models utilize partially observed data, denoted as $\boldsymbol{x}_{1:i-1}$, where each sample is regarded as a high-dimensional vector with observations up to the $(i-1)^{\text{th}}$ element. The model is then trained to predict potential values for the subsequent element, $\boldsymbol{x}_i$. Data generation is conducted through an iterative autoregressive process, where elements of each data vector is predicted one-by-one, starting from initial seeds. This can be represented as $\boldsymbol{x}_0 \xrightarrow{p_{\boldsymbol{\theta}}} ... \xrightarrow{p_{\boldsymbol{\theta}}} \boldsymbol{x}_{1:i-1} \xrightarrow{p_{\boldsymbol{\theta}}} \boldsymbol{x}_{1:i} \xrightarrow{p_{\boldsymbol{\theta}}} ... \xrightarrow{p_{\boldsymbol{\theta}}} \boldsymbol{x}_{1:d}$. The component subject to training is the autoregressive model itself. Its parameters, denoted by $\boldsymbol{\theta}$, are optimized to best predict the next elements in the sequence based on previously observed values.

As illustrated in Figure 3, all these models require real data or derived quantities (such as latent variables) as inputs to the generator during the training phase. This necessitates a significant difference in the DP training of these models compared to implicit density models, which only need indirect data access through the backward pass. In the context of typical explicit density models, DP constraints must be accounted for, given the access to real data in both the forward and backward pass. This typically results in privacy barriers being directly integrated into the update process of the generator module, as further discussed in Section 4.4.

### 3.2.3 Extensions

Our diagram has been consciously designed to encompass future developments, including potential hybrid variants of generative models. It facilitates systematic analysis of the modifications required to transition the original training pipeline to a privacy preserving one. Specifically, to train a DP variant of such a model, one could follow the following steps: (1) Illustrate the model components and information flows using diagrams analogous to those shown in Figure 3. (2) Determine the component(s) that will be provided with DP guarantees, taking into account practical use requirements and a feasible privacy-utility trade-off. (3) Establish the privacy barrier to ensure the privacy of the targeted component, which will later be made accessible for potential threat exposure. This step should consider all access paths between the target component and the data source. (4) Calculate and bound the sensitivity. (5) Implement the DP mechanism and calculate the accumulated privacy cost of the entire training process.

## 4  Taxonomy

Accompanied by a comprehensive diagram encapsulating the complete spectrum of potential design choices for deep generative models, we put forth a classification system for current DP generative methods. This system is predicated on the positioning of the privacy barrier within the diagram (Figure 1). Specifically, for explanatory purposes, we consider the key components within our diagram (the `Generator`, `Synthetic data`, `Measurement`, and `Real data`), resulting in following options for positioning the privacy barrier:

- **B1**: Between `Real data` and `Measurement`
- **B2**: Within `Measurement`
- **B3**: Between `Measurement` and `Synthetic data`
- **B4**: Within `Generator`

**B1** through **B4** are introduced sequentially, demonstrating the systematic transition of the privacy barrier from the real data source towards the generator end. The data-processing theorem 2.1 ensures that the DP guarantee is upheld as long as the data is "sanitized" through a DP mechanism prior to exposure to potential

adversaries. In this context, if a DP training algorithm safeguards against threats introduced by **B1**, then it also provides the same protective guarantee against attackers defined by **B2** through **B4**.

The generator end typically represents the smallest unit necessary for preserving the full functionality of the model, implying that the privacy barrier cannot be shifted further without compromising the operational capabilities of the generative model. Moreover, we reserve a more detailed discussion on the threat model (privacy barrier) integrated within the adopted DP mechanism (not specifically relevant to generative models) for later sections, where individual approaches will be introduced.

### 4.1 B1: Between Real Data and Measurement

**Threat Model.** Establishing a privacy barrier between the `Real data` and the `Measurement` entails using a DP mechanism to directly sanitize the data (features), thereby obtaining statistics that characterize the real data distribution for subsequent operations like computing the loss $\mathcal{L}$ as a `Measurement` that serves as the training objective for the generator. This approach provides protection against attackers who might gain access to the sanitized data features or any resultant statistics derived from the sanitized features, such as the loss measured on the sanitized data, any gradient vectors for updating the generator, and the generator's model parameters.

**General Formulation.** Methods within this category typically adopt the distribution matching framework (illustrated in Figure 3(b)), which aims to minimize the statistical distance between real and synthetic data distributions (Harder et al., 2021; Rakotomamonjy & Liva, 2021; Vinaroz et al., 2022). This distance is assessed with a static, unlearnable function, typically applying a data-independent feature extraction function $\psi$ to project the data samples into a lower-dimensional embedding space and subsequently calculating the (Euclidean) distance between the resulting embeddings of real and synthetic data. The generator is optimized to reduce the disparity between the mean embeddings of synthetic and real data, which can be interpreted as minimizing the maximum mean discrepancy (MMD) between the real and synthetic data distributions (Bińkowski et al., 2018; Li et al., 2017).

During DP training of these models, data points $\boldsymbol{x}_i$ or feature vectors $\psi(\boldsymbol{x}_i)$ are first clipped or normalized (by norm) to ensure bounded sensitivity. Subsequently, random noise is injected into the mean features derived from the real samples, e.g., via Gaussian mechanism (Definition 2.4). The objectives can be formulated as follows:

$$\text{Non-private:} \quad \min_{\boldsymbol{\theta}} \left\| \frac{1}{|\mathcal{D}|} \sum_{i=1}^{|\mathcal{D}|} \psi(\boldsymbol{x}_i) - \frac{1}{|\mathcal{S}|} \sum_{i=1}^{|\mathcal{S}|} \psi(G_{\boldsymbol{\theta}}(\boldsymbol{z}_i)) \right\|_2^2 = \min_{\boldsymbol{\theta}} \left\| \widehat{\boldsymbol{\mu}}_{\mathcal{D}} - \widehat{\boldsymbol{\mu}}_{\mathcal{S}} \right\|_2^2 \tag{3}$$

$$\text{DP:} \quad \min_{\boldsymbol{\theta}} \left\| \widetilde{\boldsymbol{\mu}}_{\mathcal{D}} - \widehat{\boldsymbol{\mu}}_{\mathcal{S}} \right\|_2^2 \quad \text{with} \quad \widetilde{\boldsymbol{\mu}}_{\mathcal{D}} = \widehat{\boldsymbol{\mu}}_{\mathcal{D}} + \mathcal{N}(0, \Delta_{\widehat{\boldsymbol{\mu}}_{\mathcal{D}}}^2 \sigma^2 \boldsymbol{I}) \tag{4}$$

with $\widehat{\boldsymbol{\mu}}_{\mathcal{D}} = \frac{1}{|\mathcal{D}|} \sum_{i=1}^{|\mathcal{D}|} \psi(\boldsymbol{x}_i)$ and $\widehat{\boldsymbol{\mu}}_{\mathcal{S}} = \frac{1}{|\mathcal{S}|} \sum_{i=1}^{|\mathcal{S}|} \psi(G_{\boldsymbol{\theta}}(\boldsymbol{z}_i))$ representing the mean features of the real and synthetic data, respectively. Meanwhile, $\widetilde{\boldsymbol{\mu}}_{\mathcal{D}}$ denotes the DP-sanitized mean real data embedding with $\Delta_{\widehat{\boldsymbol{\mu}}_{\mathcal{D}}}^2$ being the sensitivity value that characterizes the influence of each real data point on the mean embedding. A visual illustration can be found in Figure 4.

**Representative Methods.** While all methods in this category adhere to the same general formulation, they primarily diverge in their construction of the feature extraction function $\psi$ and the objective function that forms the training loss $\mathcal{L}$ for the generator. **DP-Merf** (Harder et al., 2021) employs the MMD minimization approach, optimizing a generator to minimize the difference between synthetic and real data embeddings, using random Fourier features (Rahimi & Recht, 2007) for the embedding function $\psi$. **DP-SWD** (Rakotomamonjy & Liva, 2021) instead employs random projections $\boldsymbol{u} \in \mathbb{S}^{d-1}$ for feature extraction. Specifically, DP-SWD uniformly samples $k$ random directions for data projection, thereby enabling tractable computation of one-dimensional Wasserstein distances along each projection direction. The Sliced Wasserstein Distance (SWD) (Rabin et al., 2012; Bonneel et al., 2015), which is determined as the mean of one-dimensional Wasserstein distances over DP-sanitized projections, serves as the training objective for the generator. Similar

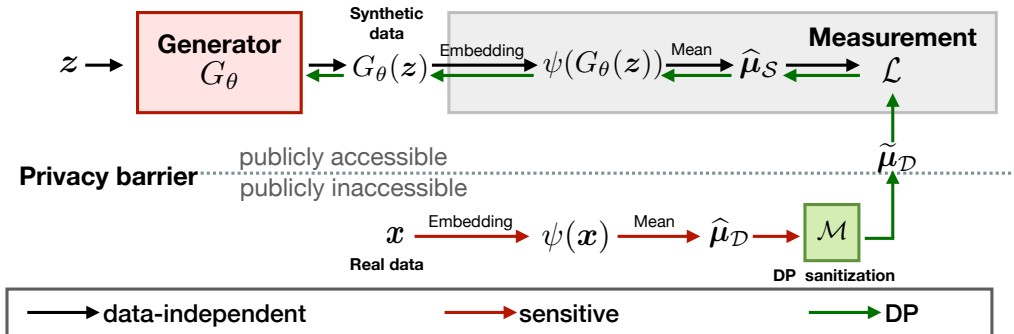

**Figure 4:** Diagram illustrating the general training procedure of methods under **B1**.

to DP-Merf, **PEARL** (Liew et al., 2021) employs the Fourier transform as the feature extraction function while offering an alternative interpretation of describing the data distribution using the characteristic function with the characteristic function distance as the objective. Furthermore, PEARL proposes learning a re-weighting function for the embedding features, placing greater emphasis on the discriminative features, in order to enhance the expressiveness of the plain Fourier features employed in the DP-Merf approach.

Recent research efforts have primarily focused on identifying informative features that can efficiently capture the underlying characteristics of the data distribution. Specifically, **DP-HP** (Vinaroz et al., 2022) employs Hermite polynomials as the feature embedding function. This choice of embedding function reduces the required feature dimension, which consequently decreases the effective sensitivity of the data mean embedding and leads to an improved signal-to-noise ratio in the DP training. **Harder et al. (2022)** further propose utilizing feature extraction layers from pre-trained classification networks that capture general concepts learned on large-scale public datasets. Additionally, **DP-NTK** (Yang et al., 2023) introduces the use of the Neural Tangent Kernel (NTK) to represent data, resulting in the gradient of the neural network function serving as the feature map, i.e., $\psi(\boldsymbol{x}) = \nabla_{\boldsymbol{\theta}} f(\boldsymbol{x}; \boldsymbol{\theta})$.

**Privacy Analysis.** The privacy analysis for methods in this category involves computing the sensitivity and applying the privacy analysis of associated noise mechanisms, such as the Gaussian mechanism (Definition 2.4). The sensitivity represents the maximum effect of an individual data point on the mean embedding:

$$\Delta^2 = \max_{\mathcal{D}, \mathcal{D}'} \|\widehat{\boldsymbol{\mu}}_{\mathcal{D}} - \widehat{\boldsymbol{\mu}}_{\mathcal{D}'}\|_2 = \Big\|\frac{1}{|\mathcal{D}|} \sum_{i=1}^{|\mathcal{D}|} \psi(\boldsymbol{x}_i) - \frac{1}{|\mathcal{D}'|} \sum_{i=1}^{|\mathcal{D}'|} \psi(\boldsymbol{x}_i')\Big\|_2 \tag{5}$$

In existing literature, the replace-one privacy notion is commonly used to compute the sensitivity value $\Delta^2$, resulting in an upper bound of $\frac{2}{|\mathcal{D}|}$ when the feature vector by construction has a norm equal to 1 or is normalized with a maximum norm of 1, i.e., $\|\psi(\boldsymbol{x})\|_2 \le 1$. Deriving the sensitivity value for the add-or-remove-one notion is slightly more technically involved, but applying existing techniques used for the replace-one notion leads to a conservative bound of $\frac{2}{|\mathcal{D}|+1}$ (See Appendix). This implies two things: first, the sensitivity value decreases inversely proportional to the size of the dataset, showing the beneficial effect of the "mean" operation over large datasets, which smooths out individual effects through population aggregation. Second, there is a minor difference in the computed sensitivity between the two privacy notions: $\frac{2}{|\mathcal{D}|+1}$ versus $\frac{2}{|\mathcal{D}|}$. This means that the current comparison results hold with negligible effect when the dataset size is sufficiently large. While achieving a tighter bound for the sensitivity value is possible with the add-or-remove-one privacy notion, it may require additional assumptions.

In contrast to other studies that compute the (worst-case) global sensitivity (Definition 2.3), the sensitivity in DP-SWD represents a form of expected value, accompanied by a sufficiently small failure probability. This efficiently harnesses the characteristics of random projections to achieve a tight sensitivity bound, but requires careful comparison to other methods. When combining this sensitivity definition with mechanisms that offer $(\varepsilon, \delta)$-DP (i.e., the relaxed DP notion), the final privacy guarantee will be weaker than $(\varepsilon, \delta)$, due to the additional failure probability derived from the sensitivity itself.

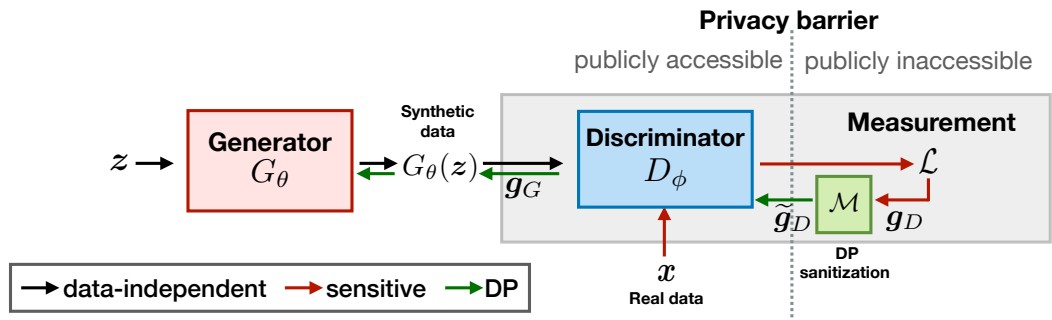

**Figure 5:** Diagram illustrating the training pipeline of **DP-GAN** with a (vertical) privacy barrier of type **B2** as shown.

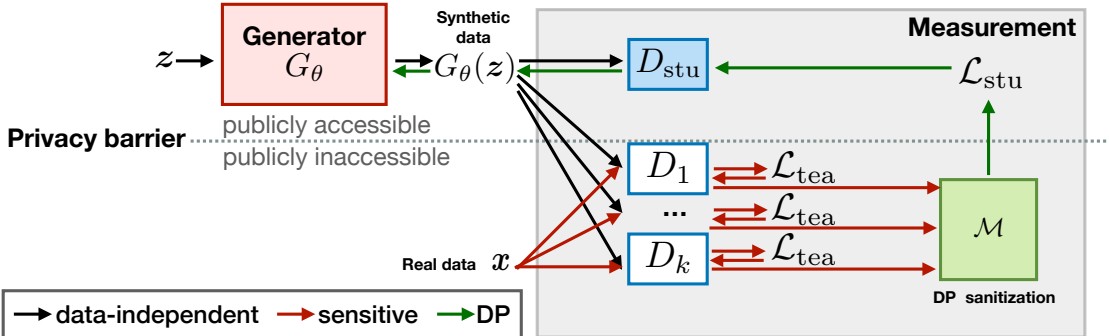

**Figure 6:** Diagram illustrating the training pipeline of **PATE-GAN** with a (horizontal) privacy barrier of type **B2** as shown. $\mathcal{L}_{\text{stu}}$ and $\mathcal{L}_{\text{tea}}$ denote the student and teacher training losses respectively, while $D_{\text{stu}}$ is the student discriminator and $D_1, ..., D_k$ represent the teacher discriminators.

**Analysis, Insights, Implications.** Methods under this category present several strengths. Firstly, the "mean" operation adopted during the extraction of descriptive feature embeddings significantly reduces the impact of each individual. This leads to a lower sensitivity value that scales in inverse proportion to the number of data points being aggregated through the "mean" operation. As a result, a strong privacy guarantee can be ensured with less randomness required from the DP mechanism. Moreover, they are straightforward to implement, typically necessitating just one instance of sanitization on the computed mean feature (known as "one-shot sanitization") throughout the training process, which further saves the privacy consumption in comparison to iterative methods. These methods also converge quickly and can yield acceptable results even under a low privacy budget, given the ease of fitting the static target, i.e., the noisy mean.

Nevertheless, they come with certain drawbacks. The static feature might not be sufficiently discriminative or informative, lacking the expressiveness found in methods that employ trainable models as critics. Furthermore, the "mean" operation could potentially induce unintended mode collapse in the generated distributions, trading off generation diversity for privacy protection. This situation warrants attention in future works, particularly in optimizing the trade-off between the expressiveness of the feature extraction method in the critic and the privacy cost of achieving such expressiveness. A promising direction could be to exploit knowledge from public non-sensitive data and/or pre-trained models that better describe data without compromising the privacy of the sensitive data.

### 4.2 B2: Within Measurement

**Threat Model.** The previous category focuses on a static, sanitized statistical summary, derived from a data-independent function, as a replacement for real data when training generative models. However, learnable functions that are able to adapt to diverse data distribution may offer superior expressive power. In this regard, a logical strategy is to incorporate DP into the measurement process, particularly by training a DP critic. This privacy barrier sits "within `Measurement`" and safeguards against adversaries with access to

the critic and subsequent quantities, including information flows to the generator. If gradient sanitization techniques like DP-SGD are employed for updating the critic, the DP mechanism further protects against attacks targeting all intermediate gradients w.r.t. the critic's parameters during the training phase.

**General Formulation.** Methods in this category follow two main principles: Firstly, they use a learnable critic (feature extraction function) that dynamically adapts to the private dataset, necessitating a boundary on the potential privacy leakage of such critic. Secondly, the generator is prohibited from accessing private real data directly, its access limited to indirect interaction through the backward pass. This ensures the generator's update signals are fully derived from the learnable critic. As such, developing a DP critic is sufficient to assure DP for the generator module (and the entire model) for privacy-preserving generation. GAN models (depicted in Figure 3(a)) meet these criteria and serve as a foundational framework that most existing DP methods in this category generally conform to.

**Representative Methods.** The implementation of the privacy barrier within the `Measurement` block is exemplified in **DP-GAN** (Zhang et al., 2018; Xie et al., 2018) and concurrent studies (Beaulieu-Jones et al., 2017; Triastcyn & Faltings, 2018; Alzantot & Srivastava, 2019; Xu et al., 2019; Torkzadehmahani et al., 2019; Frigerio et al., 2019). In this context, the discriminator, acting as the learnable critic model, is trained via DP-SGD (Section 2.1.1). The privacy of the generator is ensured by the post-processing theorem. As per the public timestamp of paper releases, this approach can be traced back to Beaulieu-Jones et al. (2017), who proposed training an ACGAN (Auxiliary Classifier GAN) (Odena et al., 2017) in a DP manner to conditionally generate samples for downstream analysis tasks on medical data. The training pipeline can be formalized as follows, with the illustration shown in Figure 5:

$$\boldsymbol{g}_D^{(t)} = \nabla_{\boldsymbol{\phi}} \mathcal{L}(G_{\boldsymbol{\theta}}, D_{\boldsymbol{\phi}}) \qquad \text{(Discriminator gradient)} \tag{6}$$

$$\boldsymbol{g}_G^{(t)} = \nabla_{\boldsymbol{\theta}} \mathcal{L}(G_{\boldsymbol{\theta}}, D_{\boldsymbol{\phi}}) \qquad \text{(Generator gradient)} \tag{7}$$

$$\widetilde{\boldsymbol{g}}_D^{(t)} = \mathcal{M}_{\sigma,C}(\boldsymbol{g}_D^{(t)}) = \text{clip}(\boldsymbol{g}_D^{(t)}, C) + \mathcal{N}(0, \sigma^2 C^2 \boldsymbol{I}) \qquad \text{(Apply DP sanitization)} \tag{8}$$

$$\boldsymbol{\phi}^{(t+1)} = \boldsymbol{\phi}^{(t)} - \eta_D \cdot \widetilde{\boldsymbol{g}}_D^{(t)} \qquad \text{(Discriminator update)} \tag{9}$$

$$\boldsymbol{\theta}^{(t+1)} = \boldsymbol{\theta}^{(t)} - \eta_G \cdot \boldsymbol{g}_G^{(t)} \qquad \text{(Generator update)} \tag{10}$$

The generator $G_{\boldsymbol{\theta}}$ and discriminator $D_{\boldsymbol{\phi}}$ are parameterized by $\boldsymbol{\theta}$ and $\boldsymbol{\phi}$, respectively, with $\eta_G$ and $\eta_D$ denoting their learning rates. $\mathcal{M}_{\sigma,C}$ refers to the Gaussian mechanism in DP-SGD, with $\sigma$ representing the noise scale and $C$ indicating the gradient clipping bound. Although we have omitted the sample index in the above equations for the sake of brevity, it should be noted that the clipping function in Equation 8 is expected to take per-example gradients as inputs, adhering to the standard procedure of DP-SGD (Section 2.1.1). Specifically, it suffices to apply the sanitization only to the gradients that depend on the real data samples, including indirect usage of real samples, such as through gradient penalty terms (Gulrajani et al., 2017).

Unlike DP-GAN that employs DP-SGD for training the DP discriminator, **PATE-GAN** (Yoon et al., 2019) leverages the PATE framework (Section 2.1.2) to train its DP (student) discriminator. PATE-GAN comprises three main components that are jointly trained throughout the process: multiple (non-private) teacher discriminators, a DP student discriminator, and a DP generator. Similar to the original PATE framework, PATE-GAN starts by partitioning the real dataset into disjoint subsets, which subsequently serve to train the teacher discriminators independently. In each training iteration, PATE-GAN follows a sequence of steps: (1) independently updating the teacher discriminators using mini-batch samples from real data partitions and synthetic samples drawn from the generator; (2) querying the teacher discriminators with a set of synthetic samples; (3) the teacher discriminators then engage in a voting process on the real/fake predictions for the synthetic samples they have received, and apply DP noise to the results of the vote; (4) training the student discriminator with the query synthetic samples as input and the DP aggregation of teacher predictions as the label; (5) finally, jointly updating the generator and the student discriminator, with the generator querying the student discriminator with new synthetic samples and obtaining update gradient signals from the DP student discriminator. A visual illustration is presented in Figure 6.

While the discriminator in the GAN framework aims to distinguish between two distributions, recent research uncovered intriguing results when the learnable critic is designed to target specific downstream tasks, such as

classification. Specifically, **Private-Set** (Chen et al., 2022a) employs a classification network as a learnable feature extractor, which is trained with DP-SGD. This learnable feature extractor, combined with the alignment in the gradients serving as the critic, encourages the synthetic data to emulate the training trajectories of the real data during the training process within a classification network, making the synthetic data useful for training downstream classifiers and safe for public release due to the DP guarantees embedded within the measurement process.

**Privacy Analysis.** Methods in this category inherit the privacy notion and sensitivity computation from their respective framework for training the DP critic (See Section 2.1.1-Section 2.1.2), while also inheriting the need for careful consideration regarding the application of data-dependent privacy analysis or adherence to privacy notion constraints to ensure comparable results. For methods grounded by DP-SGD, this results in a noticeable disparity between the replace-one and add-or-remove-one DP notions, as illustrated by the doubled sensitivity value when transitioning from the default add-or-remove-one to the replace-one notion, i.e., $C$ versus $2C$ with $C$ denoting the gradient clipping bound. Consequently, a doubled noise scale is required to achieve an ostensibly identical privacy guarantee, inevitably resulting in utility degradation and unfavorable comparison outcomes under the replace-one notion.

**Analysis, Insights, Implications.** While this training paradigm enjoys several advantages, such as ease of implementation and representative features for characterizing the difference between distributions, several challenges persist when applying such a paradigm in practice. Firstly, the joint training of a generator alongside a critic, which typically necessitates an adversarial approach, is inherently unstable due to the difficulty in maintaining equilibrium between these two components. This instability can be further amplified by the incorporation of gradient clipping and noise addition operations introduced by DP-SGD, or the additional fitting process involved in transferring knowledge from the teacher discriminators to the student one through the PATE framework. Moreover, the DP training of the critic often impedes its convergence, resulting in a sub-optimal critic that may not effectively guide the generator.

Recent studies have investigated various strategies to alleviate these challenges, particularly in the context of GANs. These include warm-starting the GAN discriminator by pre-training on public data (Zhang et al., 2018), dynamically adjusting the gradient clipping bounds during the training process (Zhang et al., 2018), re-balancing the discriminator and generator updates to restore parity to a discriminator weakened by DP noise (Bie et al., 2023), and exploiting public pre-trained GANs while restricting private modeling to the latent space (Chen et al., 2021). In the Private-Set (Chen et al., 2022a) framework that optimizes for downstream classification task, it is reported that optimizing the generator in a point-wise manner (as discussed in Section 3) or directly optimizing the synthetic set instead of the generator model can empirically lead to faster convergence and preferable when strong privacy guarantee is required. In this regard, we anticipate promising outcomes from the future development of new variants of DP-compliant training pipelines and objectives that offer improved convergence and, consequently, enhanced privacy guarantees.

### 4.3 B3: Between Measurement and Synthetic Data

**Threat Model.** In response to challenges associated with training the DP critic (Section 4.2), recent studies have proposed shifting the privacy focus from the `Measurement` to the sanitization of the intermediate signal that backpropagates to update the generator, i.e., between `Measurement` and `Synthetic data`. The goal is to preserve the critic's training stability and its utility for accurately comparing synthetic and real data, thereby guiding the generator's training effectively. This strategy ensures privacy when revealing sanitized intermediate gradients exchanged between the generator and the critic during the backward pass, as well as guarantees DP for the generator, which is updated with sanitized gradients. However, this scheme does not provide privacy guarantees for the release of the critics, since their training is conducted non-privately.

**General Formulation.** Similar to the case outlined in Section 4.2, the backbone generative models for this category are typically implicit density models. This restriction is in place as these models do not invoke direct interaction between the real data and the generator during the forward pass, which means that sanitizing the intermediate signals transmitted between the `Measurement` and `Synthetic data` is sufficient for ensuring

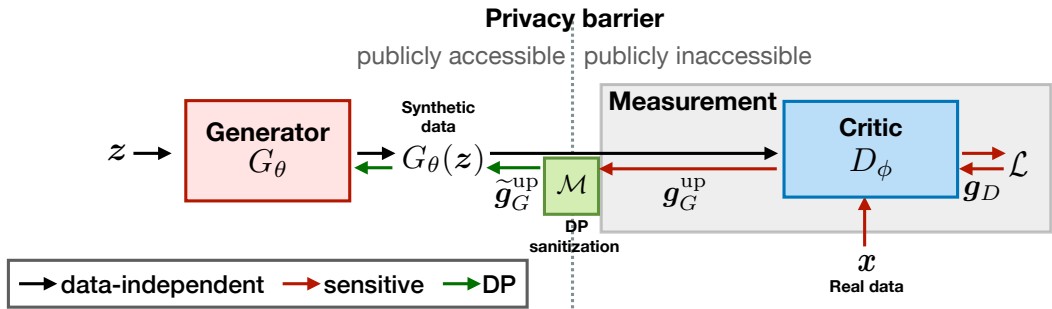

**Figure 7:** The diagram illustrates the general training process of methods incorporating the privacy barrier **B3**. In the figure, $\boldsymbol{g}_G^{\mathrm{up}}$ and $\widetilde{\boldsymbol{g}}_G^{\mathrm{up}}$ denote the upstream gradient (referenced as $\boldsymbol{g}_G^{\mathrm{upstream}}$ in Equation 12) and its sanitized variant, respectively. Note that variations exist in the formulation of critics and their corresponding training paradigms.

privacy protection. Methods in this category adhere to the gradient sanitization scheme, which introduces a DP perturbation into the gradients communicated between the critic and generator during the backward pass. This can be formulated as follows:

$$\boldsymbol{g}_G^{(t)} = \nabla_{\boldsymbol{\theta}} \mathcal{L}_G(\boldsymbol{\theta}^{(t)}) = \nabla_{G_{\boldsymbol{\theta}}(\boldsymbol{z})} \mathcal{L}_G(\boldsymbol{\theta}^{(t)}) \cdot \boldsymbol{J}_{\boldsymbol{\theta}} G_{\boldsymbol{\theta}}(\boldsymbol{z}) \tag{11}$$

$$\widetilde{\boldsymbol{g}}_G^{(t)} = \mathcal{M}\big( \underbrace{\nabla_{G_{\boldsymbol{\theta}}(\boldsymbol{z})} \mathcal{L}_G(\boldsymbol{\theta}^{(t)})}_{\boldsymbol{g}_G^{\mathrm{upstream}}} \big) \cdot \underbrace{\boldsymbol{J}_{\boldsymbol{\theta}} G_{\boldsymbol{\theta}}(\boldsymbol{z})}_{\boldsymbol{J}_G^{\mathrm{local}}} \tag{12}$$

Here, $\mathcal{L}_G$ represents the generator's loss (originating from a critic), and $\mathcal{M}$ denotes a potential DP sanitization mechanism on $\boldsymbol{g}_G^{\mathrm{upstream}}$—the gradient information backpropagating from the critic to the generator. This can be considered as the gradient of the objective with respect to the current synthetic samples. It is important to note that the second term ($\boldsymbol{J}_G^{\mathrm{local}}$), i.e., the local generator Jacobian, is computed independently of training data and thus does not require sanitization. The generator is subsequently updated with the DP sanitized gradient, i.e., $\boldsymbol{\theta}^{(t+1)} = \boldsymbol{\theta}^{(t)} - \eta_G \cdot \widetilde{\boldsymbol{g}}_G^{(t)}$. Meanwhile, the critic, if learnable, is updated normally (non-privately). A visual illustration is presented in Figure 7.

**Representative Methods.** Existing methods explored various choices for the critic and different DP mechanisms to sanitize the upstream gradients $\boldsymbol{g}_G^{\mathrm{upstream}}$. **GS-WGAN** (Chen et al., 2020a) adopts the Gaussian mechanism for sanitization and capitalizes on the inherent bounding of the gradient norm. This follows from the Lipschitz property when employing the Wasserstein distance with gradient penalty (Arjovsky et al., 2017; Gulrajani et al., 2017) as the objective when training a GAN. In contrast, **G-PATE** (Long et al., 2021) incorporates the PATE framework as its sanitization mechanism. This approach discretizes the gradients and allows multiple teacher discriminator models to vote on these discretized gradient values. The DP noisy argmax is then transferred to the generator. **DataLens** (Wang et al., 2021) further improves the signal-to-noise ratio in the PATE sanitization by employing top-K dimension compression.

In a different vein, **DP-Sinkhorn** (Cao et al., 2021) presents compelling results using a nonparametric critic. Specifically, DP-Sinkhorn estimates the Sinkhorn divergence grounded on $\ell_1$ and $\ell_2$ losses in the data space, adhering to the distribution matching generative framework as depicted in Figure 3(b). This use of a data-independent critic contributes stability to the training process and capitalizes on the privacy enhancement brought by subsampling.

**Privacy Analysis.** The privacy analysis for this method category largely aligns with the established unit sanitization mechanisms, denoted as $\mathcal{M}$, which function on upstream gradients $\boldsymbol{g}_G^{\mathrm{upstream}}$. Nevertheless, specific attention is necessary given that these intermediate gradients do not directly originate from real data samples. This scenario noticeably influences the sensitivity computation, defined formally by:

$$\Delta^2 = \max_{\mathcal{D}, \mathcal{D}'} \| f(\boldsymbol{g}_G^{\mathrm{upstream}}) - f(\boldsymbol{g}_G'^{\,\mathrm{upstream}}) \|_2 \tag{13}$$

In this equation, $f$ encapsulates the operations required to set bounds on the sensitivity and to render the associated sanitization mechanism applicable. $\boldsymbol{g}_G^{\text{upstream}}$ and $\boldsymbol{g}_G'^{\text{upstream}}$ symbolize the intermediate upstream gradients originating from neighboring datasets $\mathcal{D}$ and $\mathcal{D}'$ respectively. Specifically, $f$ performs distinct roles according to the method employed: For GS-WGAN and DP-Sinkhorn, $f$ signifies the operation of norm clipping; In G-PATE, $f$ encompasses the processes of dimension reduction and gradient discretization, and the computation of teacher voting histograms based on these discretized gradients; In the context of DataLens, rather than employing random projection and discretization as in G-PATE, $f$ adopts a top-$k$ stochastic sign quantization of the gradients. Subsequent to this operation, the teacher voting histograms are also calculated.

A direct application of the triangle inequality reveals that $\Delta^2$ equals $2C$ (with $C$ representing the gradient clipping bound) in both GS-WGAN and DP-Sinkhorn for both the replace-one and add-or-remove-one notions, while $C$ is further guaranteed to be 1 in GS-WGAN by the nature of the adopted Wasserstein objective. This is notably different from the substantial disparity between the two privacy notions in the standard DP-SGD framework. In G-PATE, the voting histogram diverges by a maximum of 2 entries for each gradient dimension, which are processed independently via DP aggregation. As for the DataLens approch, the change of one data point will at most reverse all the signs of the top-$k$ elements of gradients originated from one teacher model, leading to $\Delta^2 = 2\sqrt{k}$ (See Appendix for details).

Typically, the total privacy cost is calculated based on the RDP accountant (Theorem 2.2). Notably, each synthetic sample in a mini-batch constitutes one execution of the sanitization mechanism for the DP-SGD framework, or one query in the PATE framework. In other words, performing an update step with a mini-batch of synthetic samples on the generator can be regarded as a composition of *batch size* times its unit sanitization mechanism.

**Analysis, Insights, Implications.** Compared to previous categories (Section 4.1-4.2), shifting the privacy barrier away from the `Measurement` process itself offers several benefits. These include: (1) the flexibility to employ a powerful critic, thereby effectively guiding the generator towards capturing the characteristics of the data distribution; (2) seamless support for different privacy notions (as discussed in privacy analysis above); (3) practically simpler to properly bound the sensitivity. This can be achieved by exploiting the intrinsic properties of the objective (Chen et al., 2020a), or through the usage of the PATE framework (Long et al., 2021; Wang et al., 2021). This is particularly beneficial when compared to the previous scenario of learnable critics that typically necessitate a laborious and fragile hyperparameter search for a reasonable gradient clipping bound.

However, the increased expressive capacity comes with the trade-off of relatively high privacy consumption. The accumulation of privacy cost across iterations is notably faster in this scenario than in standard DP-SGD training of a single model: each DP update on the generator in this category equates to a *batch size* number of calls to the Gaussian mechanism, possibly without the advantage of subsampling, as detailed in the preceding privacy analysis section. This markedly contrasts with the standard DP-SGD training on a single discriminator, as mentioned in the previous category (refer to Section 4.2), where each individual DP gradient update equates to a *single* execution of the (subsampled) Gaussian mechanism.

Fortunately, this drawback has been partially mitigated through the use of data-dependent privacy analysis (as demonstrated in PATE-based methods like G-PATE and DataLens) that provides analytically tighter results that lead to stronger DP guarantees, or a data-independent critic (as in DP-Sinkhorn) that offers smooth compatibility with subsampling and better convergence. Looking forward, we anticipate further developments from refining this training paradigm, particularly through the utilization of strong backbone discriminators (and generators) trained on external non-private data, thereby optimizing privacy consumption.

### 4.4 B4: Within Generator

**Threat Model.** DP can be directly integrated into the training or deployment of a generator, the minimal unit within the generative models pipeline essential for maintaining the full generation functionality for future use. Generally, the privacy barrier safeguards against attackers who have access to the trained generator model while a more fine-grained distinguishment between the type of access (e.g., white-box or black-box) may be required depending on the application scenarios and the adopted DP mechanism. If the gradient

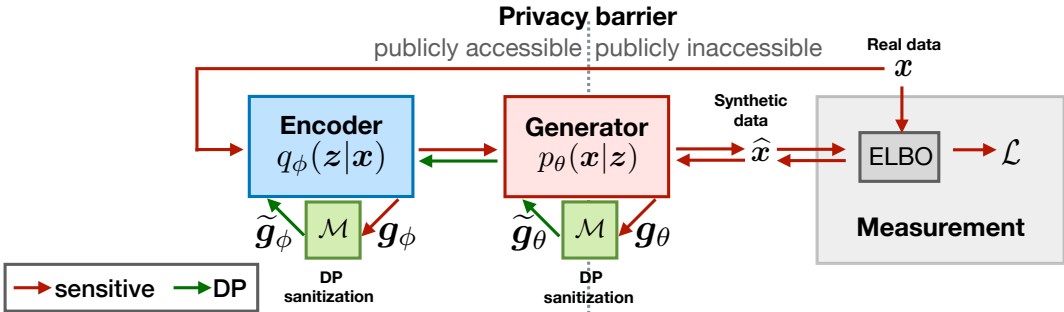

**Figure 8:** Diagram showcasing the DP training of a VAE (Section 3.2.2). This representation is also applicable to other DP model training scenarios conforming to privacy barrier **B4**, e.g., by replacing the trainable encoder with a non-trainable module.

sanitization scheme is adopted, it can protect against adversaries who can access the white-box generator (and possibly other trainable components subject to DP sanitization) and the intermediate sanitized gradients during the whole training process.

**General Formulation.** In this context, the training pipeline can be generally simplified to the standard process of training DP classification models. This process, as exemplified by the commonly used DP-SGD framework, entails bounding sensitivity through gradient clipping and subsequently injecting randomness into the generator's gradients. In contrast to category **B3**, where the upstream gradient $g_G^{\text{upstream}}$ undergoes sanitization, in this case, it is the final generator gradient $g_G^{(t)}$ (refer to Equation 11) that is being sanitized. This results in a difference equivalent to the multiplicator of the local generator Jacobian (refer to Equation 12). Special attention should be paid when implementing DP-SGD here, as additional model components (e.g., the encoder in a VAE) alongside the generator could compromise the transparency of the privacy analysis. It is crucial to ensure that the gradient clipping operation is executed accurately to effectively limit each individual real sample's influence on the generator. The presence of an additional model component may disperse individual effects across multiple gradients within a mini-batch, rendering standard per-example gradient clipping inadequate (refer to the discussion in the privacy analysis below). Moreover, to optimize model utility, it is necessary to precisely define the scope of gradient clipping and perturbation to ensure that the implementation does not introduce unnecessary noise exceeding the desired privacy guarantee.

**Representative Methods.** Existing works have realized such privacy barrier for various types of generative models, particularly those within the explicit density category. Examples include **DP Normalizing Flow** (Waites & Cummings, 2021; Jiang & Sun, 2023), **DP VAE** (Chen et al., 2018; Acs et al., 2018; Abay et al., 2019; Takagi et al., 2021; Pfitzner & Arnrich, 2022), **DP Diffusion models** (Dockhorn et al., 2022; Ghalebikesabi et al., 2023; Lyu et al., 2023), and DP training of **language models** (McMahan et al., 2018; Li et al., 2022; Mattern et al., 2022; Yue et al., 2023), which collectively illustrate the extensive potential of DP generators across numerous applications such as natural language generation, density estimation, high-quality image generation, training downstream models, and model selection. In particular, Ghalebikesabi et al. (2023) highlighted that certain training techniques advantageous for DP classification models (De et al., 2022), such as pre-training, utilization of large batch sizes, and augmentation multiplicity (Fort et al., 2021; De et al., 2022), also show effectiveness when applied to training DP generators in diffusion models. Furthermore, the work by Jiang & Sun (2023) underscores the potential efficacy of training a DP Flow model within a compressed, lower-dimensional latent space. This strategy not only circumvents the substantial computational demands (Rombach et al., 2022), but also synergizes well with DP protocols, given the direct correlation between the DP noise-to-signal-ratio and the model's dimensionality.

**Privacy Analysis.** The privacy analysis follows from the adopted DP mechanism for training the generators, similar to the standard case of training a DP classifier. A key consideration lies in the correct implementation and analysis of the privacy cost when the models comprise multiple trainable components, such as the encoder and decoder in the VAE. In such cases, simply incorporating the DP-SGD into the generator module

and conducting a standard privacy accountant is inappropriate. This is due to the fact that each training example's influence is assimilated into the encoder's parameters. Consequently, every training example, even those absent from the current mini-batch, can affect all latent variables (which serve as inputs to the generator/decoder) in each iteration, rendering the per-example gradient clipping itself insufficient for bounding the sensitivity. A proper implementation would require either enforcing DP also on the encoder (i.e., applying DP-SGD on both the encoder and decoder) or factoring this into the privacy cost computation (i.e., the DP-SGD step on the decoder should be counted as full batch Gaussian mechanism instead of a subsampled one). Moreover, in situations where each sample in a mini-batch is used more than once, such as their use over multiple time steps when training diffusion models, the cost must be accounted for every such occurrence. To deal with this, one can refer to the *multiplicity* technique (Fort et al., 2021; De et al., 2022; Ghalebikesabi et al., 2023; Dockhorn et al., 2022), which averages all gradients resulting from each unique training sample before clipping them.

**Analysis, Insights, Implications.** Methods in this category are generally easy to implement, particularly for models with only a generator as the learnable component. This reduces training to the standard classification cases, demonstrating significant potential and achieving state-of-the-art generation quality when adapted to the latest generative modeling techniques (Dockhorn et al., 2022; Ghalebikesabi et al., 2023). However, this privacy barrier setting may not be fully compatible with models containing multiple trainable components. The reason for this lies in the potential integration of training samples' effects into the parameters of components other than the generator (e.g., the encoder in VAEs, the discriminator in GANs), which substantially complicates the implementation of DP mechanisms and may lead to unexpectedly high privacy consumption. Moreover, DP methods are bounded by the expressive capability of the underlying generative model. Particularly in this category, which predominantly relies on explicit density models, the usage of simple critics (like static $\ell_1$ or $\ell_2$ loss functions) tends to restrict the capture of fine details, often delivering less desirable outcomes compared to trainable critics. For instance, VAEs have commonly produced blurrier images, whereas GANs pioneered the production of high-resolution photorealistic generations. While recent advancements in explicit density models have significantly improved their capabilities, particularly through innovative designs that enable training on extensive datasets, there is a potential limitation concerning their practical utility. This limitation primarily arises from the substantial need for sensitive training data, which is essential to achieve a satisfactory performance level with the resulting DP model in real-world applications. Looking forward, we envision future advancement on balancing the data efficiency and generation performance could largely improve the practicability of the DP methods under this category.

# 5 Discussion

## 5.1 Connection to Related Fields

While the data generation methods investigated in this work are mostly designed to capture the entire data distribution for general purposes, intriguing results are observed when the generator is intentionally guided towards enhancing its downstream utility for specific target tasks such as training neural network classifiers (Chen et al., 2022a) and answering linear queries (Liu et al., 2021). This can be achieved by employing objectives tailored for downstream tasks, rather than relying solely on general distribution divergence measures. If downstream tasks can be executed on a specific set of samples and do not require a complete understanding of the distribution, problem complexity can be further reduced by directly optimizing the synthetic samples instead of the generative models. This strategy, which trade-off the generality of general-purpose generative modeling for downstream utility, might be particularly beneficial considering the high complexity inherent to DP generation. Moreover, such framework naturally aligns with broader fields such as coreset generation, private query release, private Bayesian inference. In these scenarios, a set of synthetic data can be optimized to resemble real data for specific tasks (Wang et al., 2018; Bachem et al., 2017), substitute real data for answering queries to conserve the privacy budget under DP (Hardt & Rothblum, 2010; Hardt et al., 2012), or support privacy-preserving computation of the posterior distribution (Manousakas et al., 2020; Savitsky et al., 2022).

## 5.2 Relation to Other Summary Papers

Several related summary papers complement our work by focusing on different aspects. For instance, Tao et al. (2021) benchmark multiple DP models for tabular data; Fan (2020) and Cai et al. (2021) discuss early DP GANs; Jordon et al. (2022) and De Cristofaro (2023) provide high-level overviews of DP synthetic data generation for non-expert audiences; Hu et al. (2023) covers broad classes of DP data generation methods without focusing on the technical part of deep generative modeling; Lastly, Ponomareva et al. (2023) offer a comprehensive summary of developing and deploying general DP ML models, supplementing our focus on the technical aspects of DP generative modeling.

## 5.3 Challenges and Future Directions

**Public Knowledge.** A promising future direction which holds significant practical relevance is the exploitation of public data/knowledge in training DP generative models. Recent studies have demonstrated promising improvements in DP generation introduced by leveraging public data (Chen et al., 2021; Liu et al., 2021; Harder et al., 2022; Lyu et al., 2023) and reported high-quality generation (Ghalebikesabi et al., 2023; Lin et al., 2023) with the aid of such resources. A prevalent method for leveraging public knowledge involves utilizing large foundation models, initially pre-trained on public datasets, and subsequently fine-tuned to align with private data distributions for various applications. This approach is particularly relevant in the field of natural language processing (NLP), where the widespread availability of foundation models and the typically significant semantic overlap between public and private data renders DP fine-tuning relatively effective (Li et al., 2022). Additionally, the rapid growth of efficient fine-tuning techniques also show great potential for facilitating DP learning (Yu et al., 2021; Duan et al., 2023). While these advancements are particularly notable in the NLP domain, exploring the specific benefits and most effective strategies for applying these techniques to other data modalities is a topic that warrants further research. Furthermore, challenges that are generally associated with private learning on public data (Tramèr et al., 2022) call for further investigation. In particular, the unique difficulties specific to generative modeling, such as a small tolerance for distribution shift between the public and private data distributions, warrant additional exploration.

**Task-specific Generation.** There exists a principled trade-off between the flexibility offered by general-purpose generative modeling and the utility of task-specific data generation. In particular, capturing a complete high-dimensional data distribution is a difficult task. This task becomes even harder when considering the privacy constraints, thus making the models highly data-demanding and almost impossible for DP model to achieve reasonable performance in practice. It has also been recently questioned to what extend a well-performing general-purpose DP generative model can be realized at all (Stadler et al., 2022; Stadler & Troncoso, 2022). While it is difficult to predict how these trade-off develop in the future, task-specific (or task-guided) data generation can greatly relax the objectives, leading to real-world useful DP synthetic data (see examples discussed in Section 5.1). On the other hand, such task-specific generation is particularly advantageous for scenarios where the synthetic data is intentionally designed to be useful only for specific (benign) tasks, thereby preventing potential unauthorized data misuse.

**Conditional Generation.** While the formulas presented throughout Section 4 are illustrated through unconditional generation for simplicity and clarity, in practice, DP generation is typically executed in a conditional manner, whereby samples are generated given specific input conditions. Although implementing conditional generation is technically straightforward for all generative network backbones (Mirza & Osindero, 2014; Sohn et al., 2015; Odena et al., 2017; Winkler et al., 2019), it might necessitate additional consideration with respect to the privacy analysis. For instance, when modeling the class-conditional data feature distribution, an additional privacy budget may be allocated to learn the class label occurrence ratio for addressing class imbalance (Harder et al., 2021), contrasting with other methodologies that typically employ a data-independent uniform class-label distribution. Moreover, certain situations necessitate meticulous investigation into privacy implications and performance. Firstly, when the training process employs conditional (e.g., per-label class) sampling, additional consideration for privacy cost is imperative, as this contradicts the requirements of random sub-sampling incorporated in standard privacy cost computations. Secondly, some generative modules may integrate such conditional information in non-trivial ways (e.g., being embedded into

the module parameters beyond mere gradients (Karras et al., 2020)). This integration can mean that the conditional input might no longer be protected under DP guarantees via a vanilla DP sanitization scheme. These scenarios necessitate further exploration to ensure the reliability of privacy protections and to facilitate the development of more effective utility-preserving DP generative models.

**Federated Learning.** DP data generation models have also shown promising potential in applications related to federated training (Augenstein et al., 2019; Xin et al., 2020; Zhang et al., 2021; Triastcyn & Faltings, 2020), facilitating tasks such as privacy-preserving data inspection and debugging that were previously infeasible due to privacy constraints. Specifically, Augenstein et al. (2019) incorporated DP-SGD into the training of a GAN in a federated setting, where each client maintains a local GAN model and communicates the gradients to the server during each communication round, with the gradients being sanitized under DP noise. Moreover, Chen et al. (2020a) illustrated that the privacy barrier **B3** (Section 4.3) is seamlessly compatible with the federated training setting. In this context, only the upstream gradient (Equation 12) needs to be communicated, offering additional benefits such as improved communication efficiency. More recently, task-specific DP generation has proven particularly advantageous in alleviating non-iid challenges and enhancing convergence speed for federated learning (Xiong et al., 2023; Wang et al., 2023). Although these approaches might still require a substantial amount of client local data and computational resources, the future development of efficient algorithms is anticipated to yield fruitful outcomes.

**Evaluation and Auditing.** Evaluating generative models has historically posed a significant challenge (Lucic et al., 2018), and the same holds true for DP generation methods. While evaluating them based on specific downstream tasks has been a common approach in existing literature, it has become evident that relying solely on a single metric may be inadequate. This limitation arises from the general lack of alignment among various aspects, including downstream utility, statistical properties, and visual appearance (Alaa et al., 2022; Stadler & Troncoso, 2022; Chen et al., 2022a; Ganev et al., 2022). Consequently, there arises a need for future investigations into comprehensive metrics that consider mixed objectives to more effectively address a wide range of potential practical applications.

Furthermore, assessing the privacy guarantees of DP generators against real-world attacks (i.e., "auditing" (Jagielski et al., 2020; Nasr et al., 2023)), and quantifying the privacy risk associated with synthetic data (Stadler & Troncoso, 2022; Houssiau et al., 2022), presents a particularly intricate challenge for generative models. This complexity primarily arises from two key factors. Firstly, the measurement of privacy risks often conflicts with the primary objective of maximum likelihood, which aims to precisely fit the training data. While achieving an exact alignment with the training data aligns with training objectives, it raises a debatable question about compromising privacy protection. Deciding whether an exact match should be regarded as a privacy breach in such cases remains a matter of debate. Secondly, generative models typically exhibit low sensitivity to privacy attacks (Hayes et al., 2017; Chen et al., 2020b), which diminishes the informativeness of computed auditing scores. These challenges highlight the need for dedicated design tailored to the auditing of DP generative models.

## 6 Conclusion

In summary, we introduce a unified view coupled with a novel taxonomy that effectively characterizes existing approaches in DP deep generative modeling. Our taxonomy, which encompasses critical aspects such as threat models, general formulation, detailed descriptions, privacy analysis, as well as insights and broader implications, provides a consolidated platform for systematically exploring potential innovative methodologies while leveraging the strengths of existing techniques. Furthermore, we present a comprehensive introduction to the core principles of DP and generative modeling, accompanied by substantial insights and discussions regarding essential considerations for future research in this area.

## Acknowledgements

This work is partially funded by the Helmholtz Association within the project "Privatsphären-schützende Analytik in der Medizin" (PrivateAIM, 01ZZ2316G), the BMBF with the project "Represantative, synthetische

Gesundheitsdaten mit starken Privatspharengarantien" (PriSyn, 16KISAO29K) and ELSA – European Lighthouse on Secure and Safe AI, funded by the European Union under grant agreement No. 101070617. Views and opinions expressed are however those of the authors only and do not necessarily reflect those of the European Union or European Commission. Neither the European Union nor the European Commission can be held responsible for them. Dingfan Chen acknowledges the funding from the Qualcomm Innovation Fellowship Europe. We acknowledge Matthias Reisser for providing valuable comments on this work.

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

# Appendix

## A    Summary of Existing Works

| Approach | Privacy barrier | Privacy notion | Sensitivity type | Generative framework | DP framework | Code |
|---|---|---|---|---|---|---|
| DP-Merf (Harder et al., 2021) | **B1** | Replace-one | Global | Distribution matching | Gaussian | [1] |
| DP-SWD (Rakotomamonjy & Liva, 2021) | **B1** | Replace-one | Smooth | Distribution matching | Gaussian | [2] |
| PEARL (Liew et al., 2021) | **B1** | Replace-one | Global | Distribution matching | Gaussian | [3] |
| DP-HP Vinaroz et al. (2022) | **B1** | Replace-one | Global | Distribution matching | Gaussian | [4] |
| DP-GEN (Chen et al., 2022b) | **∼B1** | – | – | Energy-based model | – | [5] |
| Harder et al. (2022) | **B1** | Replace-one | Global | Distribution matching | Gaussian | [6] |
| DP-NTK (Yang et al., 2023) | **B1** | Replace-one | Global | Distribution matching | Gaussian | [7] |
| DPSDA (Lin et al., 2023) | **B1** | Add-or-remove-one | Global | Diffusion | Gaussian | [8] |
| SPRINT-gan (Beaulieu-Jones et al., 2017) | **B2** | Add-or-remove-one | Global | GAN | DP-SGD | [9] |
| dp-GAN (Zhang et al., 2018) | **B2** | Add-or-remove-one | Global | GAN | DP-SGD | [10] |
| DPGAN Xie et al. (2018) | **B2** | Add-or-remove-one | Global | GAN | DP-SGD | [11] |
| Triastcyn & Faltings (2018) | **B2** | Add-or-remove-one | – | GAN | empirical DP | – |
| PATE-GAN (Yoon et al., 2019) | **B2** | Both | Local | GAN | PATE | [12] |
| Alzantot & Srivastava (2019) | **B2** | Add-or-remove-one | Global | GAN | DP-SGD | [13] |
| Xu et al. (2019) | **B2** | Add-or-remove-one | Global | GAN | DP-SGD | – |
| DP-CGAN (Torkzadehmahani et al., 2019) | **B2** | Add-or-remove-one | Global | GAN | DP-SGD | [14] |
| Frigerio et al. (2019) | **B2** | Add-or-remove-one | Global | GAN | DP-SGD | [15] |
| DPMI (Chen et al., 2021) | **B2** | Add-or-remove-one | Global | GAN | DP-SGD | – |
| DPautoGAN (Tantipongpipat et al., 2021) | **B2** | Add-or-remove-one | Global | GAN | DP-SGD | [16] |
| Private-Set (Chen et al., 2022a) | **B2** | Add-or-remove-one | Global | Distribution matching | DP-SGD | [17] |
| Bie et al. (2023) | **B2** | Add-or-remove-one | Global | GAN | DP-SGD | – |
| GS-WGAN (Chen et al., 2020a) | **B3** | Both | Global | GAN | DP-SGD | [18] |
| G-PATE (Long et al., 2021) | **B3** | Both | Local | GAN | PATE | [19] |
| DataLens (Wang et al., 2021) | **B3** | Both | Global/Local | GAN | PATE | [20] |
| DP-Sinkhorn (Cao et al., 2021) | **B3** | Both | Global | Distribution matching | DP-SGD | [21] |
| DP-GM (Acs et al., 2018) | **B4** | Add-or-remove-one | Global | VAE | DP-SGD | – |
| DP-VaeGM (Chen et al., 2018) | **B4** | Add-or-remove-one | Global | VAE, AE$^+$ | DP-SGD | – |
| DP-SYN (Abay et al., 2019) | **B4** | Add-or-remove-one | Global | AE$^+$ | DP-SGD | – |
| P3GM (Takagi et al., 2021) | **B4** | Add-or-remove-one | Global | VAE | DP-SGD | [22] |
| DP-NF (Waites & Cummings, 2021) | **B4** | Add-or-remove-one | Global | Flow | DP-SGD | [23] |
| DP$^2$-VAE (Jiang et al., 2022) | **B4** | Add-or-remove-one | Global | VAE | DP-SGD | – |
| DPDM (Dockhorn et al., 2022) | **B4** | Add-or-remove-one | Global | Diffusion | DP-SGD | [24] |
| Ghalebikesabi et al. (2023) | **B4** | Add-or-remove-one | Global | Diffusion | DP-SGD | – |
| DP-LDM (Lyu et al., 2023) | **B4** | Add-or-remove-one | Global | Diffusion | DP-SGD | [25] |
| DP-LFlow Jiang & Sun (2023) | **B4** | Add-or-remove-one | Global | Flow | DP-SGD | [26] |

**Table 1:** Table summary of existing works most relevant to our contributions. The shaded area corresponds to approaches that require public data features.

[1]  https://github.com/ParkLabML/DP-MERF
[2]  https://github.com/arakotom/dp_swd
[3]  https://github.com/spliew/pearl
[4]  https://github.com/parklabml/dp-hp
[5]  https://github.com/chiamuyu/DPGEN
[6]  https://github.com/ParkLabML/DP-MERF
[7]  https://github.com/FreddieNeverLeft/DP-NTK
[8]  https://github.com/microsoft/DPSDA
[9]  https://github.com/greenelab/SPRINT_gan
[10]  https://github.com/alps-lab/dpgan
[11]  https://github.com/illidanlab/dpgan
[12]  https://github.com/vanderschaarlab/mlforhealthlabpub/tree/main/alg/pategan
[13]  https://github.com/nesl/nist_differential_privacy_synthetic_data_challenge/
[14]  https://github.com/reihaneh-torkzadehmahani/DP-CGAN
[15]  https://github.com/SAP-samples/security-research-differentially-private-generative-models
[16]  https://github.com/DPautoGAN/DPautoGAN
[17]  https://github.com/DingfanChen/Private-Set
[18]  https://github.com/DingfanChen/GS-WGAN
[19]  https://github.com/AI-secure/G-PATE
[20]  https://github.com/AI-secure/DataLens
[21]  https://github.com/nv-tlabs/DP-Sinkhorn_code
[22]  https://github.com/tkgsn/P3GM
[23]  https://github.com/ChrisWaites/jax-flows/tree/master/research/dp-flows
[24]  https://github.com/nv-tlabs/DPDM
[25]  https://github.com/SaiyueLyu/DP-LDM
[26]  https://github.com/dihjiang/DP-LFlow

Note that the DP training of language models typically falls within DP generative modeling and can be categorized as the "**within generator**" category in our taxonomy. However, the training strategies adopted are more akin to the DP training of general (discriminative) models, which is not the primary focus of our work. Hence, these approaches are not exhaustively mentioned in the table above.

## B   Additional Notes on Potential Methods with Privacy Barrier B1

In the DP deep generative modeling literature, existing approaches with privacy barrier between `Real data` and `Measurement`  (Section 4.1) typically release sanitized features in a *condensed and aggregated* form. In this sense, recent approaches, which may deviate from the general "mean embedding" formulation (as shown in Equation 3-4), but still publish a sanitized statistical summary of the private dataset, such as **DPSDA** (Lin et al., 2023), fall into this category. Specifically, **DPSDA** sanitizes a count histogram that summarizes the distribution of real data and employs it as a measurement to refine the synthetic data distribution, thereby rendering it more similar to the real private data distribution.

However, one might wonder if it is feasible to release a DP database in the *original* form of the real data, *prior to* the training of a generative model. A positive example of this idea can be found in the Small Database Mechanism (**SmallDB**) in the context of private query release, introduced in Section 4.1 of Dwork et al. (2014). This mechanism outputs a sanitized database in the same form as the original data, by selecting the database (from all possible sets of the data universe) via the exponential mechanism with a utility function of the negative error to the query release problem (difference in the query answer on the synthetic versus the real database). However, as the name suggests, the use of such an algorithm is largely limited to small (low-dimensional) datasets. This is mainly due to the exponential growth of the data universe with dimensionality, which drastically increases the computational burden and undermines the accuracy guarantees.

While **DP-GEN** (Chen et al., 2022b) attempted to apply a similar idea to deep generative models, the output space of their generation method only supports (has non-zero probability) combinations of its input *private* dataset (See detailed proofs in Appendix B of Dockhorn et al. (2022)), instead of the entire data universe. This invalidates their claimed privacy guarantee, and the performance of a proper implementation of such a "direct database release" approach on high-dimensional data remains unclear.

## C   Additional Sensitivity Analysis

### C.1   Privacy barrier B1

**Sensitivity of DP-Merf (Harder et al., 2021) and the General Formulation in Section 4.1.**   It can be clearly seen that the $\ell_2$-sensitivity for the replace-one notion is $\frac{2}{m}$, where $m = |\mathcal{D}|$ represents the size of the private dataset, as demonstrated in the original paper. Subsequently, we proceed to derive a conservative bound for the sensitivity value in the DP-Merf method under the add-or-remove-one DP notion, which can be generalized to other approaches within the same category (Section 4.1), including Liew et al. (2021); Vinaroz et al. (2022); Harder et al. (2022); Yang et al. (2023). For the add-one case, we let $m = |\mathcal{D}|$ and assume,

without loss of generality, that $\mathcal{D}' = \mathcal{D} \cup \{\boldsymbol{x}'_{m+1}\}$ and $\boldsymbol{x}'_i = \boldsymbol{x}_i$ for all $i = 1, ..., m$.

$$
\begin{aligned}
\Delta^2 &= \max_{\mathcal{D},\mathcal{D}'} \left\| \frac{1}{m+1} \sum_{i=1}^{m+1} \phi(\boldsymbol{x}'_i) - \frac{1}{m} \sum_{i=1}^{m} \phi(\boldsymbol{x}_i) \right\|_2 \\
&= \max_{\boldsymbol{x}'_{m+1},\boldsymbol{A}} \left\| \frac{1}{m+1}(\phi(\boldsymbol{x}'_{m+1}) + \boldsymbol{A}) - \frac{1}{m}\boldsymbol{A} \right\|_2 \\
&= \max_{\boldsymbol{x}'_{m+1},\boldsymbol{A}} \left\| \frac{1}{(m+1)m}\boldsymbol{A} - \frac{1}{m+1}\phi(\boldsymbol{x}'_{m+1}) \right\|_2 \\
&\leq \max_{\boldsymbol{A}} \left\| \frac{1}{(m+1)m}\boldsymbol{A} \right\|_2 + \max_{\boldsymbol{x}'_{m+1}} \left\| \frac{1}{m+1}\phi(\boldsymbol{x}'_{m+1}) \right\|_2 \\
&\leq \frac{1}{(m+1)m}m + \frac{1}{m+1} = \frac{2}{m+1}
\end{aligned}
$$

where $\boldsymbol{A} = \sum_{i=1}^{m} \phi(\boldsymbol{x}_i)$ for brevity. The inequalities follow from the triangle inequality and the fact that $\|\phi(\cdot)\|_2 = 1$

Similarly, for the remove-one case, we let $m = |\mathcal{D}|$, $\mathcal{D}' \cup \{\boldsymbol{x}_m\} = \mathcal{D}$ and $\boldsymbol{x}'_i = \boldsymbol{x}_i$ for all $i = 1, ..., m-1$.

$$
\begin{aligned}
\Delta^2 &= \max_{\mathcal{D},\mathcal{D}'} \left\| \frac{1}{m-1} \sum_{i=1}^{m-1} \phi(\boldsymbol{x}'_i) - \frac{1}{m} \sum_{i=1}^{m} \phi(\boldsymbol{x}_i) \right\|_2 \\
&= \max_{\boldsymbol{x}_m,\boldsymbol{A}} \left\| \frac{1}{m-1}\boldsymbol{A} - \frac{1}{m}(\boldsymbol{A} + \phi(\boldsymbol{x}_m)) \right\|_2 \\
&= \max_{\boldsymbol{x}_m,\boldsymbol{A}} \left\| \frac{1}{(m-1)m}\boldsymbol{A} - \frac{1}{m}\phi(\boldsymbol{x}_m) \right\|_2 \\
&\leq \max_{\boldsymbol{A}} \left\| \frac{1}{(m-1)m}\boldsymbol{A} \right\|_2 + \max_{\boldsymbol{x}_m} \left\| \frac{1}{m}\phi(\boldsymbol{x}_m) \right\|_2 \\
&\leq \frac{1}{(m-1)m}(m-1) + \frac{1}{m} = \frac{2}{m}
\end{aligned}
$$

with $\boldsymbol{A} = \sum_{i=1}^{m-1} \phi(\boldsymbol{x}_i)$. The inequalities follow from the triangle inequality and the fact that $\|\phi(\cdot)\|_2 = 1$

**Sensitivity of DP-SWD (Rakotomamonjy & Liva, 2021).** The sensitivity is calculated as the maximum difference over two embeddings, determined after performing random projections on two neighboring datasets. The "replace-one" notion is adopted to simplify the analysis. With a probability of at least $1 - \delta$, it can be shown that:

$$\|\boldsymbol{X}\boldsymbol{U} - \boldsymbol{X}'\boldsymbol{U}\|_F^2 \leq w(k, \delta)$$

with $w(k, \delta) = \frac{k}{d} + \frac{2}{3} \ln \frac{1}{\delta} + \frac{2}{d}\sqrt{k\frac{d-1}{d+2} \ln \frac{1}{\delta}}$. Here $\boldsymbol{X}, \boldsymbol{X}'$ denote data matrices in $\mathbb{R}^{|\mathcal{D}| \times d}$ for neighboring datasets $\mathcal{D}, \mathcal{D}'$ under the bounded-DP notion, while $\boldsymbol{U} \in \mathbb{R}^{d \times k}$ represents the random projection matrix with each column independently drawn from $\mathbb{S}^{d-1}$. Additionally, it is ensured that $\|\boldsymbol{X}_{i,:} - \boldsymbol{X}'_{i,:}\|_2 \leq 1$ for all $i$ by pre-processing the dataset, making each sample record have unit norm. To prove the desired result, the sensitivity is first transformed into a summation of $k$ i.i.d random variables following the beta distribution $B(1/2, (d-1)/2)$, which then allows the application of Bernstein's inequality to establish concentration bounds for the summation. For a more detailed proof, please refer to Appendix 8.1-8.2 in Rakotomamonjy & Liva (2021).

**Sensitivity of DPSDA (Lin et al., 2023).** The core component of DPSDA is the method of constructing a nearest neighbors histogram that describes the real data distribution while providing DP guarantees (refer to Algorithm 2 in Lin et al. (2023)). Specifically, for every real sample $\boldsymbol{x}_i$ in the private dataset $\mathcal{D}$, the algorithm identifies its nearest synthetic counterparts and constructs a histogram. This histogram represents the frequency of each existing synthetic sample $\boldsymbol{s}_k$ being the closest to the real samples. Given a synthetic

dataset consisting of $n$ samples $\{s_k\}_{k=1}^n$ and let $m = |\mathcal{D}|$:

$$h_j = \left| i : i \in [m], j = \underset{k \in [n]}{\arg\min}\, d(\boldsymbol{x}_i, \boldsymbol{s}_k) \right| \quad \text{for } j = 1, ..., n$$

where $\boldsymbol{h} = (h_1, ..., h_n)$ builds up the histogram with each $h_j$ reflecting the number of real samples for which the corresponding synthetic sample $\boldsymbol{s}_j$ is the nearest neighbor, based on the distance metric $d$. Subsequently, DP Gaussian noise is added to the histogram for providing privacy guarantees: $\boldsymbol{h} = \boldsymbol{h} + \mathcal{N}(0, \sigma\boldsymbol{I})$.

For the add-or-remove-one notion, we can assume that w.l.o.g. the neighboring datasets $\mathcal{D}, \mathcal{D}'$ satisfy $\mathcal{D}' \cup \{\boldsymbol{x}_m\} = \mathcal{D}$ (or $\mathcal{D}' = \mathcal{D} \cup \{\boldsymbol{x}_m\}$). Let $\boldsymbol{s}_j$ be the closest synthetic sample to $\boldsymbol{x}_m$ and $\boldsymbol{h}, \boldsymbol{h}'$ represent the histograms on $\mathcal{D}$ and $\mathcal{D}'$ respectively. The $\ell_2$-sensitivity is then given by:

$$\begin{aligned}
\Delta^2 &= \max_{\mathcal{D}, \mathcal{D}'} \|(h_1, \cdots, h_n) - (h_1', \cdots, h_n')\|_2 \\
&= \max_{h_j, h_j'} \|(0, ..., 0, h_j - h_j', 0, ..., 0)\|_2 \\
&= 1
\end{aligned}$$

For the replace-one notion, we define neighboring datasets $\mathcal{D}, \mathcal{D}'$ to satisfy $\mathcal{D}' \cup \{\boldsymbol{x}_m\} = \mathcal{D} \cup \{\boldsymbol{x}_m'\}$ with $\boldsymbol{x}_m \neq \boldsymbol{x}_m'$. The $\ell_2$-sensitivity is defined by:

$$\begin{aligned}
\Delta^2 &= \max_{\mathcal{D}, \mathcal{D}'} \|(h_1, \cdots, h_n) - (h_1', \cdots, h_n')\|_2 \\
&= \max_{h_j, h_j', h_k, h_k'} \|(0, ..., 0, h_j - h_j', 0, ..., 0, h_k - h_k', 0, ..., 0)\|_2 \\
&= \sqrt{1^2 + 1^2} = \sqrt{2}
\end{aligned}$$

where $\boldsymbol{s}_j$ and $\boldsymbol{s}_k$ are the closet synthetic samples to $\boldsymbol{x}_m$ and $\boldsymbol{x}_m'$ respectively, while w.l.o.g. $j < k$.

## C.2 Privacy barrier B2

The sensitivity analysis for methods in this category inherits the approach used in the DP-SGD and the PATE framework, which is presented below.

**Sensitivity of DP-SGD (Section 2.1.1).** The main component of the DP-SGD algorithm can be formalized as follows:

$$\text{Clip: } \bar{\boldsymbol{g}}_t(\boldsymbol{x}_i) \leftarrow \boldsymbol{g}_t(\boldsymbol{x}_i) / \max\left(1, \frac{\|\boldsymbol{g}_t(\boldsymbol{x}_i)\|_2}{C}\right)$$

$$\text{Add noise: } \widetilde{\boldsymbol{g}}_t \leftarrow \frac{1}{B}\left(\sum_i \bar{\boldsymbol{g}}_t(\boldsymbol{x}_i) + \mathcal{N}(0, \sigma^2 C^2 \boldsymbol{I})\right)$$

where $\boldsymbol{g}_t(\boldsymbol{x}_i) = \nabla_{\boldsymbol{\theta}_t}\mathcal{L}(\boldsymbol{\theta}_t, \boldsymbol{x}_i)$ denotes the gradient on sample $\boldsymbol{x}_i$ at iteration $t$, $C$ represents the clipping bound, $B$ is the batch size, $\sigma$ is the noise scale, and the summation is taken over all samples in the batch. The sensitivity in DP-SGD is computed as:

$$\Delta^2 = \max_{\mathcal{D}, \mathcal{D}'} \|\sum_i \bar{\boldsymbol{g}}_t(\boldsymbol{x}_i) - \sum_i \bar{\boldsymbol{g}}_t(\boldsymbol{x}_i')\|_2$$

For the add-or-remove-one DP notion, let $\mathcal{D}', \mathcal{D}$ only differ in the existence of $\boldsymbol{x}_i'$, i.e., $\mathcal{D}' = \mathcal{D} \cup \{\boldsymbol{x}_i'\}$, it is easy to see that

$$\Delta^2 = \max_{\boldsymbol{x}_i'} \|\bar{\boldsymbol{g}}_t(\boldsymbol{x}_i')\|_2 \leq C$$

For the replace-one DP notion, w.l.o.g. let $\mathcal{D}' \cup \{\boldsymbol{x}_i'\} = \mathcal{D} \cup \{\boldsymbol{x}_i\}$, thus

$$\Delta^2 = \max_{\boldsymbol{x}_i', \boldsymbol{x}_i} \|\bar{\boldsymbol{g}}_t(\boldsymbol{x}_i) - \bar{\boldsymbol{g}}_t(\boldsymbol{x}_i')\|_2 \leq 2C$$

due to the triangle inequality.

**Sensitivity of PATE (Section 2.1.2).** Given $m$ teachers, $c$ possible label classes and an input vector $\boldsymbol{x}$, the "votes" of teachers that assign class $j$ to a query input $\bar{\boldsymbol{x}}$ is denoted as:

$$n_j(\bar{\boldsymbol{x}}) = |i : i \in [m], f_i(\bar{\boldsymbol{x}}) = j| \quad \text{for } j = 1, ..., c$$

with $f_i$ denotes the $i$-th teacher model. And the histogram of the teachers' vote histogram is:

$$\bar{n}(\bar{\boldsymbol{x}}) = (n_1, \cdots, n_c) \in \mathbb{N}^c$$

As each training data sample only influences a single teacher due to the disjoint partitioning, changing one data sample in the training dataset—whether it's removal, addition, or replacement—will at most alter the votes (by 1) for two classes, denoted here as classes $i$ and $j$, on any possible query sample $\bar{\boldsymbol{x}}$. Let the vote histograms resulting from neighboring datasets $\mathcal{D}, \mathcal{D}'$ be $(n_1, \cdots, n_c)$ and $(n'_1, \cdots, n'_c)$ respectively, the global sensitivity can be represented as:

$$
\begin{aligned}
\Delta^1 &= \max_{\mathcal{D},\mathcal{D}'} \|(n_1, \cdots, n_c) - (n'_1, \cdots, n'_c)\|_1 \\
&= \max_{n_i, n'_i, n_j, n'_j} \|(0, ..., 0, n_i - n'_i, 0, ..., 0, n_j - n'_j, 0, ..., 0)\|_1 \\
&= \max_{n_i, n'_i} |n_i - n'_i| + \max_{n_j, n'_j} |n_j - n'_j| \leq 2 \\
\Delta^2 &= \max_{n_i, n'_i, n_j, n'_j} \|(0, ..., 0, n_i - n'_i, 0, ..., 0, n_j - n'_j, 0, ..., 0)\|_2 \\
&= \max_{n_i, n'_i, n_j, n'_j} \sqrt{(n_i - n'_i)^2 + (n_j - n'_j)^2} \leq \sqrt{2}
\end{aligned}
$$

This holds for all possible query samples $\bar{\boldsymbol{x}}$.

The $\ell_1$- and $\ell_2$-sensitivities calibrate the two variants of noise mechanisms used in PATE: the Gaussian NoisyMax (GNMax) and the max-of-Laplacian (LNMax). The GNMax is defined as:

$$\text{PATE}_\sigma(\bar{\boldsymbol{x}}) = \arg\max_{j \in [c]} \{n_j(\bar{\boldsymbol{x}}) + \mathcal{N}(0, \sigma^2)\}$$

and the LNMax as:

$$\text{PATE}_\gamma(\bar{\boldsymbol{x}}) = \arg\max_{j \in [c]} \{n_j(\bar{\boldsymbol{x}}) + Lap(1/\gamma)\}$$

### C.3 Privacy barrier B3

**Sensitivity of GS-WGAN (Chen et al., 2020a) and DP-Sinkhorn (Cao et al., 2021).** The sensitivity for both GS-WGAN and DP-Sinkhorn can be derived via triangle inequality:

$$
\begin{aligned}
\Delta^2 &= \max_{\mathcal{D},\mathcal{D}'} \|f(\boldsymbol{g}_G^{\text{upstream}}) - f(\boldsymbol{g}'_G{}^{\text{upstream}})\|_2 \\
&\leq \max_{\mathcal{D}} \|f(\boldsymbol{g}_G^{\text{upstream}})\|_2 + \max_{\mathcal{D}'} \|f(\boldsymbol{g}'_G{}^{\text{upstream}})\|_2 \\
&\leq 2C
\end{aligned}
$$

with $f$ denoting the gradient clipping operation and $C$ the clipping bound. Notably, no matter which privacy notion is used, both terms ($\max_{\mathcal{D}} \|f(\boldsymbol{g}_G^{\text{upstream}})\|_2$ and $\max_{\mathcal{D}'} \|f(\boldsymbol{g}'_G{}^{\text{upstream}})\|_2$) are upper-bounded by the gradient clipping bound $C$.

**Sensitivity of DataLens (Wang et al., 2021).** Given $m$ teachers, the $d$-dimensional gradients yielded from each teacher $i$ after applying top-$k$ sign quantization take the following form (refer to Algorithm 2 in Wang et al. (2021)):

$$\hat{\boldsymbol{g}}_i \in \{0, 1, -1\}^d \quad \text{with} \quad \|\hat{\boldsymbol{g}}_i\|_1 = k \quad \text{and} \quad \|\hat{\boldsymbol{g}}_i\|_2 = \sqrt{k}$$

In other words, $\boldsymbol{g}_i$ contains exactly $k$ non-zero elements, with the non-zero elements taking values of either 1 or $-1$, depending on the sign of the original upstream gradient.

Consider gradient sets $\{\hat{\boldsymbol{g}}_i\}_{i=1}^m$ and $\{\hat{\boldsymbol{g}}_i'\}_{i=1}^m$ which originate from neighboring datasets $\mathcal{D}$ and $\mathcal{D}'$ respectively. As the influence of each data point is limited to a single teacher model, these gradient sets differ by at most one element. Without loss of generality, let's assume they diverge in the $i$-th element. The $\ell_2$-sensitivity is then computed as follows:

$$
\begin{aligned}
\Delta^2 &= \max_{\mathcal{D}, \mathcal{D}'} \Big\| \sum_{i=1}^m \hat{\boldsymbol{g}}_i - \sum_{i=1}^m \hat{\boldsymbol{g}}_i' \Big\|_2 \\
&= \max_{\hat{\boldsymbol{g}}_i, \hat{\boldsymbol{g}}_i'} \big\| \hat{\boldsymbol{g}}_i - \hat{\boldsymbol{g}}_i' \big\|_2 \\
&\leq \|\hat{\boldsymbol{g}}_i\|_2 + \|\hat{\boldsymbol{g}}_i'\|_2 = 2\sqrt{k}
\end{aligned}
$$

### C.4 Privacy barrier B4

The sensitivity analysis for methods in this category adheres to the DP-SGD framework. While special considerations may be required to ensure the implementation correctly adheres to this framework, these considerations typically do not alter the sensitivity analysis itself.

## D  Additional Background on Privacy Cost Accumulation

Theorem 2.2 (presented in Section 2) provides a straightforward method for calculating the aggregated privacy cost when composing multiple (potentially heterogeneous) DP mechanisms. In this section, we present more details regarding determining the accumulated privacy cost over multiple executions of *sampled Gaussian mechanisms* (Definition D.1).

**Definition D.1** (Sampled Gaussian Mechanism (SGM) (Abadi et al., 2016; Mironov et al., 2019)). Let $f$ be an arbitrary function mapping subsets of $\mathcal{D}$ to $\mathbb{R}^d$. The sampled Gaussian mechanism (SGM) parametrized with the sampling rate $0 < q \leq 1$ and the noise multiplier $\sigma > 0$ is defined as

$$\mathrm{SG}_{q,\sigma} \triangleq f\left(\{\boldsymbol{x} : \boldsymbol{x} \in \mathcal{D} \text{ is sampled with probability } q\}\right) + \mathcal{N}(0, \sigma^2 \boldsymbol{I}_d)$$

where each element of $\mathcal{D}$ is sampled independently at random with probability $q$ without replacement.

The sampled Gaussian mechanism consists of adding i.i.d Gaussian noise with zero mean and variance $\sigma^2$ to each coordinate of the true output of $f$, i.e., $\mathrm{SG}_{q,\sigma}$ injects random vectors from a multivariate isotropic Gaussian distribution $\mathcal{N}(0, \sigma^2 \boldsymbol{I}_d)$ and into the true output, where $\boldsymbol{I}_d$ is written as $\boldsymbol{I}$ if unambiguous in the given context.

**Theorem D.1.** (Mironov et al., 2019) Let $\mathrm{SG}_{q,\sigma}$ be the sampled Gaussian mechanism for some function $f$ with $\Delta_f^2 \leq 1$ for any adjacent $\mathcal{D}, \mathcal{D}'$ under the add-or-remove-one notion. Then $\mathrm{SG}_{q,\sigma}$ satisfies $(\alpha, \rho)$-RDP if

$$\rho \leq D_\alpha\left(\mathcal{N}(0, \sigma^2) \,\big\|\, (1-q)\mathcal{N}(0, \sigma^2) + q\mathcal{N}(1, \sigma^2)\right)$$

$$\text{and} \quad \rho \leq D_\alpha\left((1-q)\mathcal{N}(0, \sigma^2) + q\mathcal{N}(1, \sigma^2) \,\big\|\, \mathcal{N}(0, \sigma^2)\right)$$

Theorem D.1 reduce the problem of proving the RDP bound for $\mathrm{SG}_{q,\sigma}$ to a simple special case of a mixture of one-dimensional Gaussians.

**Theorem D.2.** (Mironov et al., 2019) Let $\text{SG}_{q,\sigma}$ be the sampled Gaussian mechanism for some function $f$ and under the assumption $\Delta_f^2 \leq 1$ for any adjacent $\mathcal{D}, \mathcal{D}'$ under the add-or-remove-one notion. Let $\mu_0$ denote the pdf of $\mathcal{N}(0, \sigma^2)$, $\mu_1$ denote the pdf of $\mathcal{N}(1, \sigma^2)$, and let $\mu$ be the mixture of two Gaussians $\mu = (1-q)\mu_0 + q\mu_1$. Then $\text{SG}_{q,\sigma}$ satisfies $(\alpha, \rho)$-RDP if

$$\rho \leq \frac{1}{\alpha - 1} \log\left(\max\{A_\alpha, B_\alpha\}\right)$$

where

$$A_\alpha \triangleq \mathbb{E}_{z \sim \mu_0}[(\mu(z)/\mu_0(z))^\alpha]$$
$$B_\alpha \triangleq \mathbb{E}_{z \sim \mu}[(\mu_0(z)/\mu(z))^\alpha]$$

Theorem D.2 states that applying SGM to a function of sensitivity (Equation 2.3) at most 1 satisfies $(\alpha, \rho)$-RDP if $\rho \leq \frac{1}{\alpha-1} \log(\max\{A_\alpha, B_\alpha\})$. Thus, analyzing RDP properties of SGM is equivalent to upper bounding $A_\alpha$ and $B_\alpha$.

**Corollary D.1.** (Mironov et al., 2019) $A_\alpha \geq B_\alpha$ for any $\alpha \geq 1$.

This allows reformulation of the RDP bound as

$$\rho \leq \frac{1}{\alpha - 1} \log A_\alpha$$

The $A_\alpha$ can be calculated for a range of $\alpha$ values using the numerically stable computation approach presented in Section 3.3 of Mironov et al. (2019), which is implemented in standard DP packages such as Opacus[27] and Tensorflow-privacy[28]. Then, the smallest $A_\alpha$ (tightest bound) is used to upper bound $\rho$ and later the RDP privacy cost is converted to $(\varepsilon, \delta)$-DP via Theorem 2.3. Notably, this approach generalizes previous results such as moment accountant (Abadi et al., 2016) (See Table 1 in Mironov et al. (2019) for a summary).

---

[27]  https://opacus.ai/
[28]  https://github.com/tensorflow/privacy

