# OpenReview forum: "A Unified View of Differentially Private Deep Generative Modeling"
_TMLR — Accepted by TMLR_

### Review · Reviewer_Lvy8 · 2023-12-07

**Summary Of Contributions:**

This paper provides a comprehensive review on existing approaches for differentially private deep generative modeling from a unified prespective. Different methods are systematically introduced, categorized, and analyzed on the strength and weakness. Open issues and potential future issues in the field of DP generation methods are also discussed.

**Audience:**

Yes

**Claims And Evidence:**

Yes

**Requested Changes:**

See weakness part

**Strengths And Weaknesses:**

## Strength

1. This paper is a well-written review paper for existing techniques of DP generative models. All key concepts are well introduced with high-quality figure illustrations. The organization and presentation is friendly to broader audiences
2. The paper provides systematic categorizations of existing techniques under a unified framework, clearly discussing the setting, strength, and weakness of different methods
3. The paper clearly identifies open questions in the field for future developments

## Weakness

As the motivation of DP generative model is to enable downstream statistical analysis by sharing a sanitized form of the data, the tradeoff between privacy and utility naturally come into play. Such tradeoff is an important factor for end users to select a proper DP method for their need. Though the paper made extensive discussion on the privacy guarantee of different methods, the implication on utility is not discussed. There also lacks a comprehensive discussion on potential downstream task that can be benefited from the DP generative models. Adding the discussion on downstream task and the utility tradeoff would be especially helpful for practitioners aiming to utilize the DP model.

---

### Review · Reviewer_UFro · 2023-12-17

**Summary Of Contributions:**

This paper provides a systemization of the knowledge surrounding differentially private data-synthesis. It starts off by providing primers on differential privacy and generative models, and then moves on to categorizing different DP-synthesis methods based on where the privacy barrier is placed. They provide four such groups: barrier between measure and real data, barrier within measure, barrier between measure and synthetic data and barrier within generator. They then discuss the work that falls under each category.

**Audience:**

Yes

**Claims And Evidence:**

Yes

**Requested Changes:**

1. Missing all NLP related synthesis references: The paper is missing a discussion on synthesized data in language, in particular the following papers:

[1] Mattern, Justus, et al. "Differentially Private Language Models for Secure Data Sharing." Proceedings of the 2022 Conference on Empirical Methods in Natural Language Processing. 2022.
[2] Xiang Yue, Huseyin Inan, Xuechen Li, Girish Kumar, Julia McAnallen, Hoda Shajari, Huan Sun, David Levitan, and Robert Sim. 2023. Synthetic Text Generation with Differential Privacy: A Simple and Practical Recipe. In Proceedings of the 61st Annual Meeting of the Association for Computational Linguistics (Volume 1: Long Papers)
[3]  Fatemehsadat Mireshghallah, Yu Su, Tatsunori Hashimoto, Jason Eisner, and Richard Shin. 2023. Privacy-Preserving Domain Adaptation of Semantic Parsers. In Proceedings of the 61st Annual Meeting of the Association for Computational Linguistics (Volume 1: Long Papers)

I believe all of these methods would fall under the 'within generator' category.


2. The within measure category seems a bit unintuitive for me. Barrier is usually not within a thing, it is between componenets. I think both methods (PATE-GAN and DP-GAN) would be between real-data and measure. Because for PATE, it's actually the noisy output that is fed to the measure, so the noise is not in the measure. Similar case for GAN.

**Strengths And Weaknesses:**

Strengths:
1. The visualizations, I think they are all very intuitive, stimulating and to the point. I loved them. They are also pretty accurate and thorough.
2. The flow of the paper and the categorization: I think the way the paper is written will benefit both experts and non-expert, which is not always the case for survey papers!


Weaknesses:
1. Although the paper claims to unify all DP-synthesis approaches, and to work on different modalities, the work from the NLP literature is missing. See the requested changes for a full list. In general, I think a discussion of modalities and different challenges for each one is needed.

2. The paper is more like a good tutorial, than an SoK paper, as in it doesn't really fully help the reader understand when to use what method. What is next in this field?

3. I think the within metric categorization is redundant. See below.

---

### Review · Reviewer_bWxG · 2023-12-29

**Summary Of Contributions:**

This paper provides a unified view of differential privacy for generative models. It is a synthesis paper, that aptly summarizes about 10 years of methods in the intersection of DP and Gen AI into a coherent mathematical framework.

**Audience:**

Yes

**Broader Impact Concerns:**

NA.

**Claims And Evidence:**

Yes

**Requested Changes:**

I would recommend that the authors add the following to items to improve this work: A discussion on the similarities and distinctions when applying DP as a in fine-tuning a generative model compared to in pre-training. With generative models now serving as foundation models, it would be helpful to understand the differences along this dimension.

Furthermore, the authors should probably be a bit more thorough concerning DP fine-tuning of transformer-based LLMs. This is an extremely timely topic, as these models are becoming increasingly pervasive.

I do not considering this "critical", rather, strongly recommended.

**Strengths And Weaknesses:**

The exposition is clear and thorough. I consider this a strong review paper. The novelty is limited, but I agree that the proposed framework and taxonomy is a strong contribution.

I recommend this paper for acceptance to TMLR. I would consider this paper an solid resource for both academic researchers, industry practioners, and grad students.

---

### Decision · Action_Editor_VLtS · 2024-03-08

**Recommendation:** Accept as is

**Comment:**

This submission is recommended for a survey certification on the basis of the content and recommendation of the reviewers.

**Audience:**

Deep generative modeling and privacy in ML are clear relevant topics to the TMLR audience and are topics of many recent articles in related venues.

**Claims And Evidence:**

The submission is a survey article that provides a unified mathematical presentation, and a large coverage of related papers.  The reviewers appreciated the contributions and were unanimous in their recommendation of acceptance.  Requested additional references on NLP were not included in the final version, but the relevant reviewer did not object in the final review stage.